# Elevated temperature inhibits SARS-CoV-2 replication in respiratory epithelium independently of IFN-mediated innate immune defenses

Vanessa Herder[1], Kieran Dee[1], Joanna K. Wojtus[1], Ilaria Epifano[1],
Daniel Goldfarb[1], Christoforos Rozario[1], Quan Gu[1], Ana Da Silva Filipe[1],
Kyriaki Nomikou[1], Jenna Nichols[1], Ruth F. Jarrett[1], Andrew Stevenson[1],
Steven McFarlane[1], Meredith E. Stewart[1], Agnieszka M. Szemiel[1], Rute M. Pinto[1],
Andreu Masdefiol Garriga[1,2], Chris Davis[1], Jay Allan[1], Sheila V. Graham[1]*, Pablo
R. Murcia[1]*, Chris Boutell[1]*

1 MRC-University of Glasgow Centre for Virus Research (CVR), Glasgow, Scotland United Kingdom,
2 University of Glasgow School of Veterinary Medicine, Glasgow, Scotland United Kingdom

☯ These authors contributed equally to this work.
* Sheila.Graham@glasgow.ac.uk (SVG); Pablo.Murcia@glasgow.ac.uk (PRM); chris.boutell@glasgow.ac.uk
(CB)

pbio.3001065

International de Recherche en Infectiologie (CIRI),
FRANCE

**Data Availability Statement:** The datasets
generated and analyzed during the current study

## Abstract

The pandemic spread of Severe Acute Respiratory Syndrome Coronavirus 2 (SARS-CoV-2), the etiological agent of Coronavirus Disease 2019 (COVID-19), represents an ongoing international health crisis. A key symptom of SARS-CoV-2 infection is the onset of fever, with a hyperthermic temperature range of 38 to 41°C. Fever is an evolutionarily conserved host response to microbial infection that can influence the outcome of viral pathogenicity and regulation of host innate and adaptive immune responses. However, it remains to be determined what effect elevated temperature has on SARS-CoV-2 replication. Utilizing a three-dimensional (3D) air–liquid interface (ALI) model that closely mimics the natural tissue physiology of SARS-CoV-2 infection in the respiratory airway, we identify tissue temperature to play an important role in the regulation of SARS-CoV-2 infection. Respiratory tissue incubated at 40°C remained permissive to SARS-CoV-2 entry but refractory to viral transcription, leading to significantly reduced levels of viral RNA replication and apical shedding of infectious virus. We identify tissue temperature to play an important role in the differential regulation of epithelial host responses to SARS-CoV-2 infection that impact upon multiple pathways, including intracellular immune regulation, without disruption to general transcription or epithelium integrity. We present the first evidence that febrile temperatures associated with COVID-19 inhibit SARS-CoV-2 replication in respiratory epithelia. Our data identify an important role for tissue temperature in the epithelial restriction of SARS-CoV-2 independently of canonical interferon (IFN)-mediated antiviral immune defenses.

are available in Supplemental Files; S1 to S9 data. RNA-Seq and amplicon sequencing data sets are freely available from the European Nucleotide Archive, accession number PRJEB41332.

**Funding:** VH was funded by the German Research Foundation (Deutsche Forschungsgemeinschaft; project number 406109949) and the Federal Ministry of Food and Agriculture (BMEL; Förderkennzeichen: 01KI1723G). KD and PRM were funded by the Medical Research Council (MRC; MC_UU_12014/9 to PRM). JKW was funded by an MRC CVR DTA award (MC_ST_U18018). IE was funded by a CSO project grant (TCS/19/11). DG was funded by an MRC-DTP award (MR/R502327/1). CR was funded by a BBSRC-CTP award (BB/R505341/1). QG was funded by the MRC (MC_UU_12014/12). KN and ASF were funded by the MRC (MC_UU_12018/12). MES was funded by the MRC (MC PC 19026). AMS was funded by a UKRI/DHSC grant (BB/R019843/1 to Brian Willett, MRC-UoG CVR) and MRC CoV supplement grant (MC_PC_19026). RMP was funded by the MRC (MC_UU_12014/10). AMG was funded by studentship awards from the University of Glasgow School of Veterinary Medicine (Georgina D. Gardner, 145813; John Crawford, 123939). CB and SMF were funded by the MRC (MC_UU_12014/5 to CB). The funders had no role in study design, data collection and analysis, decision to publish, or preparation of the manuscript.

**Competing interests:** The authors have declared that no competing interests exist.

**Abbreviations:** ACE2, angiotensin-converting enzyme 2; ALI, air–liquid interface; ARDS, acute respiratory distress syndrome; COVID-19, Coronavirus Disease 2019; CPM, counts per million; DAMP, damage-associated molecular pattern; DEG, differentially expressed gene; DMEM, Dulbecco's Modified Eagle Medium; FCS, fetal calf serum; FDR, false discovery rate; GLM, generalized linear model; HBEp, human bronchiolar epithelial; H&E, hematoxylin and eosin; IAV, influenza A virus; ICU, intensive care unit; IFN, interferon; IHC, immunohistochemistry; IL-6, interleukin 6; ISG, IFN-stimulated gene; ISH, in situ hybridization; JAK, Janus kinase; lncRNA, long noncoding RNA; MHV, murine hepatitis virus; MR, mapped read; PAMP, pathogen associated molecular pattern; PCA, principle component analysis; PFU, plaque-forming unit; PRR, pattern recognition receptor; RdRp, RNA-dependent RNA polymerase; RNA-Seq, RNA sequencing; RT-qPCR, reverse transcription quantitative PCR; RT, room temperature; Ruxo, Ruxolitinib; SARS-CoV-2, Severe Acute Respiratory Syndrome Coronavirus

# Introduction

The pandemic spread of severe acute respiratory syndrome coronavirus 2 (SARS-CoV-2; [1–3]) is an ongoing international health crisis with over 245 million infections and 5 million reported deaths worldwide to date (WHO; https://covid19.who.int; November 2021). The spectrum of SARS-CoV-2–related disease (Coronavirus Disease 2019 (COVID-19)) is highly variable, ranging from asymptomatic viral shedding to acute respiratory distress syndrome (ARDS), multiorgan failure, and death. Besides coughing, dyspnea, and fatigue, fever (also known as pyrexia) is one of the most frequently reported symptoms [4–8]. Fever is an evolutionarily conserved response to microbial infection and inflammation that can influence the regulation of multiple cellular processes, including host innate and adaptive immune responses [9,10]. Unlike hyperthermia or heat stroke, fever represents a controlled shift in body temperature regulation induced by the expression of exogenous (microbial) and endogenous (host) pyrogenic regulatory factors, including pathogen associated molecular patterns (PAMPs) and proinflammatory cytokines (e.g., interleukin 6 (IL-6)) [9,10]. Body temperature naturally varies throughout the day, with age, sex, and ethnic origin being contributing factors [9,10]. In healthy middle-aged adults, febrile temperatures range from 38 to 41˚C (ΔT approximately 1 to 4˚C above baseline), with low (38 to 39˚C), moderate (39.1 to 40˚C), high (40.1 to 41.1˚C), and hyperpyrexia (>41.1˚C) temperature ranges [10]. Temperature elevation is known to confer protection against a number of respiratory pathogens [9,11], with antipyretic drug treatment leading to increased mortality in intensive care unit (ICU) patients infected with influenza A virus (IAV) [12–14].

With respect to COVID-19, up to 90% of hospitalized patients show low (44%) to moderate (13% to 34%) grade fever [4–8], with ICU patients presenting a 10% higher prevalence of fever relative to non-ICU patients [6,15]. In vitro studies have shown that SARS-CoV-2 replicates more efficiently at lower temperatures associated with the upper respiratory airway (33˚C), which correlates with an overall weaker interferon (IFN)-mediated immune response to infection relative to lower respiratory airway (37˚C) infection [16]. These data suggest that tissue temperature could be a significant factor in the host immune response to SARS-CoV-2 infection and COVID-19 disease progression. However, it remains to be determined what effect elevated temperature (>37˚C) has on SARS-CoV-2 replication. We therefore set out to determine the net effect of temperature elevation on SARS-CoV-2 infection within respiratory epithelial tissue.

Utilizing a three-dimensional (3D) respiratory model that closely mimics the tissue physiology and cellular tropism of SARS-CoV-2 infection observed in the respiratory airway [16–24], we demonstrate temperature elevation (≥39˚C) to restrict the replication and propagation of SARS-CoV-2 independently of the induction of IFN-mediated antiviral immune defenses. We show that respiratory epithelium remains permissive to SARS-CoV-2 infection at temperatures up to 40˚C, but to restrict the initiation of viral transcription leading to reduced levels of viral RNA (vRNA) replication and apical shedding of infectious virus. We identify temperature to play an important role in the differential regulation of multiple epithelial host responses to SARS-CoV-2 infection, including epigenetic, long noncoding RNA (lncRNA), and immunity-related pathways. Our data identify an important role for tissue temperature in the epithelial restriction of SARS-CoV-2 replication independently of canonical IFN-mediated antiviral immune defenses previously reported to restrict SARS-CoV-2 infection.

2; sgRNA, subgenomic RNA; SNP, single-
nucleotide polymorphism; TEER, transepithelial
electrical resistance; ts, temperature-sensitive;
tsFIPV, temperature-sensitive feline infectious
peritonitis virus; vRNA, viral RNA.

# Results

## Differentiation of primary human bronchial epithelial cells into ciliated respiratory epithelium supports SARS-CoV-2 replication in vitro

To establish a model suitable to study the effect of temperature on SARS-CoV-2 replication, we differentiated primary human bronchiolar epithelial (HBEp) cells isolated from 3 independent donors into stratified respiratory epithelium. Hematoxylin and eosin (H&E) and immunohistochemistry (IHC) staining of respiratory cultures demonstrated these tissues to contain a mixture of epithelial and goblet cells, with high levels of apical ciliation and angiotensin-converting enzyme 2 (ACE2) expression, the principal surface receptor of SARS-CoV-2 (Fig 1A and 1B) [25–28]. Infection of respiratory airway cultures with SARS-CoV-2 (England/02/2020;

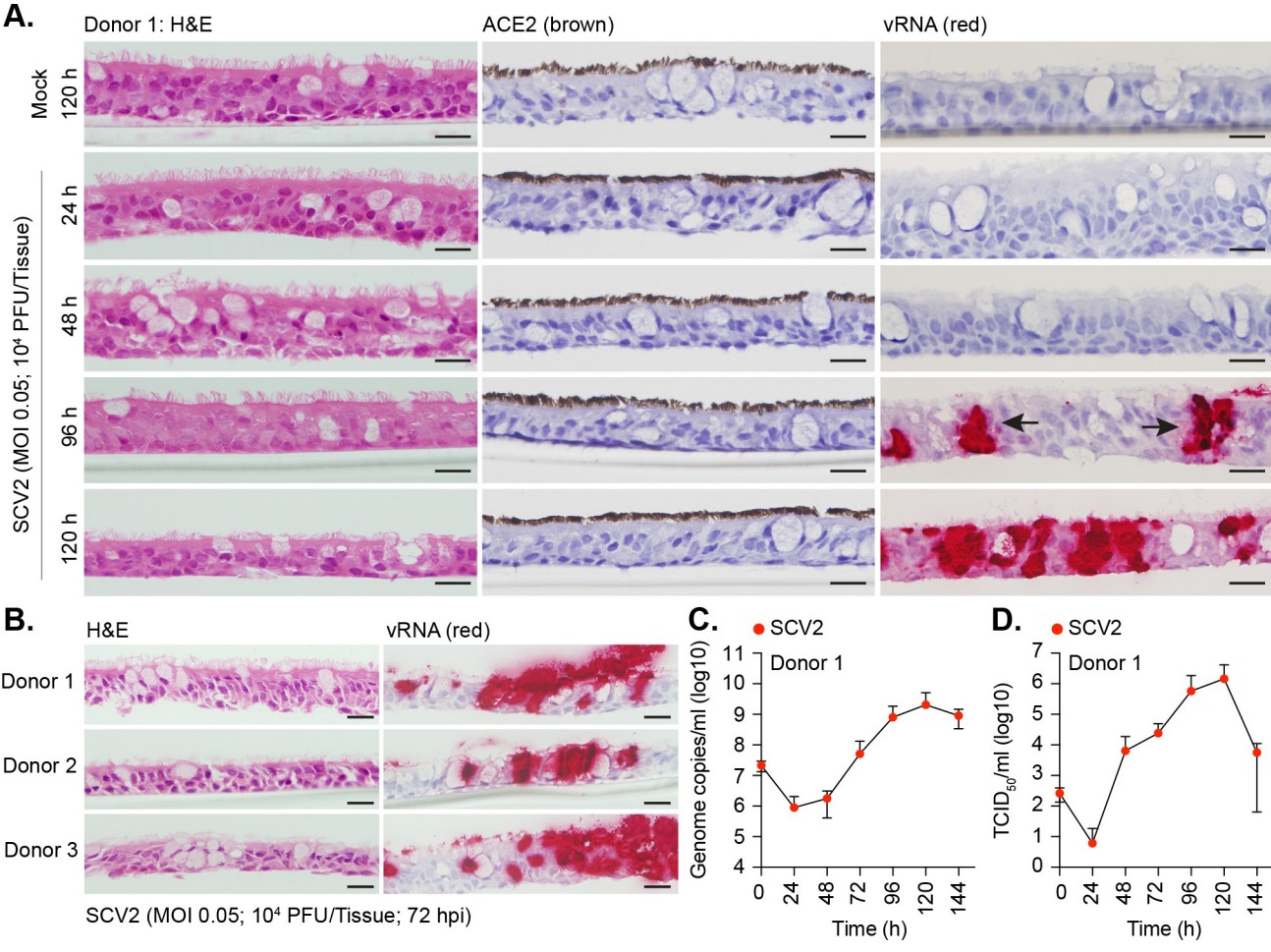

**Fig 1. Differentiation of primary bronchial epithelial airway cultures supports SARS-CoV-2 replication.** Primary bronchial epithelial cells isolated from 3 independent donors (donor 1, male Caucasian aged 63 years; donor 2, Hispanic male aged 62 years; donor 3, Caucasian female aged 16 years; all nonsmokers) were seeded onto 6.5 mm transwells and differentiated under ALI conditions for $\geq$35 days. Ciliated respiratory cultures were mock treated (media only) or SARS-CoV-2 (SCV2; MOI 0.05, $10^4$ PFU/Tissue) infected at 37˚C for the indicated times (h). (A, B) Representative images of H&E, ACE2 IHC (brown), or SARS-CoV-2 RNA ISH (red) stained sections. Hematoxylin was used as a counter stain. Scale bars = 20 μm. (C) Genome copies per ml of SARS-CoV-2 in apical washes harvested over time as determined RT-qPCR. $N \geq 7$ tissues per condition derived from a minimum of 3 independent biological experiments. Means and SD shown. (D) $TCID_{50}$ assay measuring infectious viral load in apical washes harvested from SARS-CoV-2 infected tissues over time. Means and SD shown. $N = 6$ tissues per condition derived from a minimum of 3 independent biological experiments. Raw values presented in S9 Data. ACE2, angiotensin-converting enzyme 2; ALI, air–liquid interface; H&E, hematoxylin and eosin; IHC, immunohistochemistry; ISH, in situ hybridization; PFU, plaque-forming unit; RT-qPCR, reverse transcription quantitative PCR; SARS-CoV-2, Severe Acute Respiratory Syndrome Coronavirus 2; vRNA, viral RNA.

MOI 0.05, $10^4$ plaque-forming unit (PFU)/Tissue) at 37˚C demonstrated these tissues to support infection and viral replication, with intraepithelial and apical vRNA accumulation readily detected by in situ hybridization (ISH) by 72 h postinfection (Fig 1A and 1B). Notably, we observed discrete clusters of vRNA accumulation within respiratory epithelia (Fig 1A, arrows), indicative of localized areas of intraepithelial infection, propagation, and spread [29,30]. The overall morphology of the respiratory epithelium remained largely intact, with little shedding of ciliated cells from the epithelial surface (Fig 1A, 120 h). Measurement of genome copies by reverse transcription quantitative PCR (RT-qPCR) and infectious virus by $TCID_{50}$ in apical washes collected over time demonstrated the linear phase of virus release to occur between 48 and 120 h (Fig 1C and 1D). A substantial drop in apical infectious titers was observed at 144 h postinfection (Fig 1D), consistent with multiple waves of virus release over prolonged periods of incubation in primary airway cultures [31].

## Respiratory airway cultures induce a heat stress response at elevated temperature

While the heat stress response has been extensively investigated in two-dimensional (2D) cell culture model systems [32,33], this pathway remains poorly characterized in 3D respiratory epithelia under air–liquid interface (ALI). We therefore investigated the influence of temperature on our 3D respiratory model. Mock-treated respiratory cultures were incubated at 37, 39, or 40˚C (representative of core body temperature and low to moderate grade febrile temperatures, respectively) prior to tissue fixation or RNA extraction and RNA sequencing (RNA-Seq). H&E staining and transepithelial electrical resistance (TEER) measurements demonstrated no morphological or membrane integrity changes to the respiratory epithelium upon temperature elevation (Fig 2A and 2B). RNA-Seq analysis of respiratory epithelia demonstrated no difference in the expression level of ACE2 (Fig 2C, $p = 0.6753$) or a reference set of 24 genes known to be constitutively expressed across a wide range of tissues and cell types (Fig 2D, $p = 0.2136$; [34–36]) upon temperature elevation. These data indicate that general transcription remains largely unperturbed at elevated temperatures up to 40˚C. Out of the 867 differentially expressed genes (DEGs) identified (Fig 2E; $Q < 0.05$), DEG enrichment was observed across multiple pathways in tissues incubated at 40 relative to 37˚C, including cellular response to stress, RNA polymerase I promoter opening, and DNA methylation (Fig 2F and S1 Data). Significant DEG enrichment was also observed in cellular response to heat stress (Fig 2F [arrow], 2G; S1 Data). We conclude that our respiratory model induces a significant heat stress response at elevated temperature without visible damage to the epithelium or induction of innate immune defenses that may otherwise influence the interpretation of SARS-CoV-2 replication studies at elevated temperature.

## Elevated temperature restricts the replication of SARS-CoV-2 in respiratory airway cultures

We next examined the effect of elevated temperature on SARS-CoV-2 replication. Respiratory cultures were incubated at 37, 39, or 40˚C for 24 h prior to SARS-CoV-2 infection and continued incubation at their respective temperature. Measurement of genome copies and infectious virus from apical washes collected over time (24 to 72 h) demonstrated that extracellular viral titers were significantly decreased at both 39 and 40˚C relative to 37˚C (Fig 3A and 3B). Importantly, SARS-CoV-2 infection at 37˚C for 24 h prior to temperature upshift also led to reduced genome copy titers in apical washes (Fig 3C; $p = 0.0056$). These data indicate that temperature elevation, either prior to or following SARS-CoV-2 infection, is inhibitory to viral replication and/or release of viral particles. These data identify febrile temperature ranges

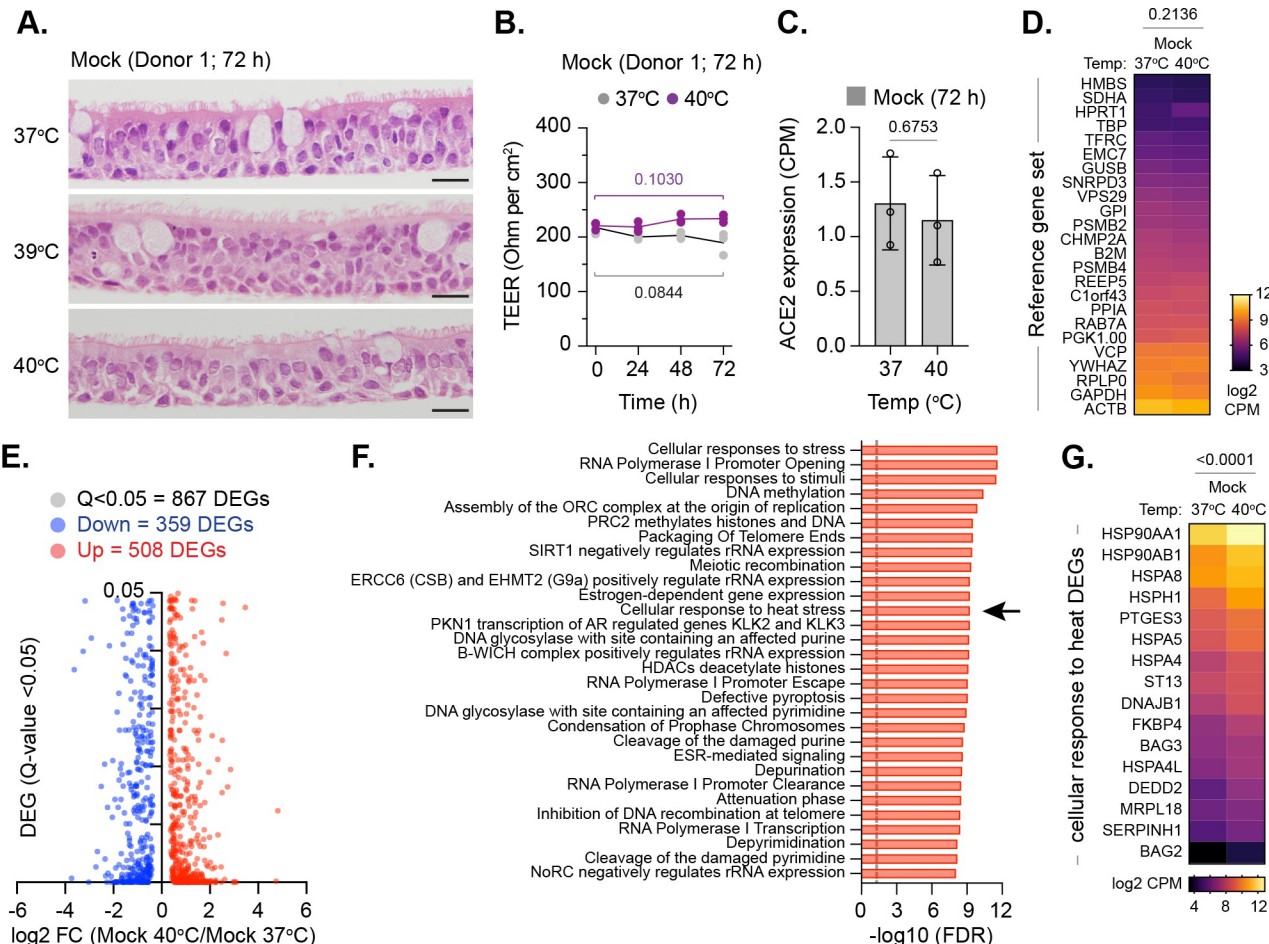

**Fig 2. Respiratory airway cultures induce a heat stress response upon incubation at elevated temperature.** Ciliated respiratory cultures differentiated from HBEp cells isolated from 3 independent donors (donor 1, male Caucasian aged 63 years; donor 2, Hispanic male aged 62 years; donor 3, Caucasian female aged 16 years; all nonsmokers) were incubated at 37, 39, or 40°C for 72 h prior to fixation or RNA extraction and RNA-Seq. (A) Representative H&E stained tissue sections (donor 1) at 72 h. Scale bars = 20 μm. (B) TEER (Ohm per cm$^2$) measurements of respiratory epithelia integrity (donor 1) at 37 or 40°C. $N$ = 3 tissues per condition; lines, mean; all data points shown; $p$-values shown, unpaired two-tailed $t$ test. (C) ACE2 expression levels (CPM) in respiratory cultures incubated at 37 or 40°C; means and SD shown; $p$-value shown, unpaired two-tailed $t$ test. (D) Expression values (log2 CPM) of a reference gene set (24 genes; [34–36]) in respiratory cultures incubated at 37 or 40°C; $p$-value shown, paired two-tailed $t$ test. (E) Scatter plots showing high confidence (Q < 0.05) DEG transcripts identified between respiratory cultures incubated at 37 or 40°C; up-regulated DEGs, red circles; down-regulated DEGs, blue circles. (F) Reactome pathway analysis of mapped up-regulated DEGs. Top 30 pathways (FDR < 0.05; plotted as −log10 FDR) shown. Dotted line, threshold of significance (−log10 FDR of 0.05). (G) Expression values (log2 CPM) of DEGs associated with cellular response to heat stress pathway (R-HSA-3371556; arrow in F); $p$-value shown, paired two-tailed $t$ test. (C to H) RNA-Seq data derived from RNA isolated from 3 donors (donors 1 to 3) per sample condition derived from 3 independent experiments per donor. Raw values presented in S1 and S9 Data. ACE2, angiotensin-converting enzyme 2; CPM, counts per million; DEG, differentially expressed gene; FDR, false discovery rate; HBEp, human bronchiolar epithelial; H&E, hematoxylin and eosin; TEER, transepithelial electrical resistance.

associated with low (39°C) to moderate (40°C) grade fever to restrict SARS-CoV-2 propagation in respiratory epithelia.

We next investigated whether this temperature-dependent restriction was donor dependent. Respiratory cultures were incubated at 37 or 40°C for 24 h prior to SARS-CoV-2 infection and continued incubation at their respective temperature. Measurement of genome copy titers in apical washes demonstrated viral titers were significantly reduced at 40°C in a donor-independent manner (Fig 4A and 4B). Notably, donor-dependent profiles of restriction were observed (Fig 4B), with respiratory tissue derived from donor 1 being the most refractory to SARS-CoV-2 replication at 40°C. We next examined if the decreased yield of infectious virus

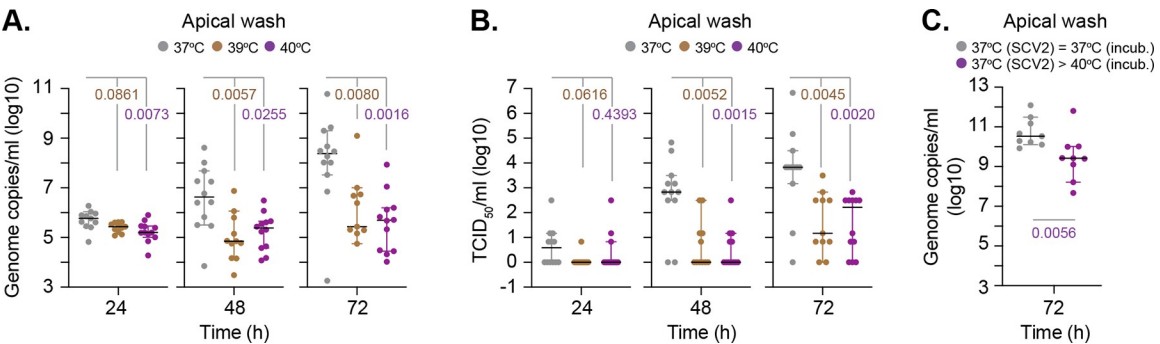

**Fig 3. Elevated temperature restricts SARS-CoV-2 replication in respiratory airway cultures.** Ciliated respiratory cultures differentiated from HBEp cells isolated from donor 1 (male Caucasian aged 63 years) were incubated at 37, 39, or 40°C for 24 h prior to SARS-CoV-2 (SCV2; MOI 0.05, $10^4$ PFU/Tissue) infection and incubation at the indicated temperatures. Apical washes were collected over time (as indicated). (A) Genome copies per ml of SARS-CoV-2 in apical washes were determined RT-qPCR. $N \geq 11$ tissues per condition derived from a minimum of 3 independent biological experiments. Black line, median; whisker, 95% confidence interval; all data points shown; p-values shown, one-way ANOVA Kruskal–Wallis test. (B) TCID$_{50}$ assay measuring infectious viral load in apical washes harvested from SARS-CoV-2–infected respiratory cultures. $N \geq 11$ tissues per condition derived from a minimum of 3 independent biological experiments. Black line, median; whisker, 95% confidence interval; all data points shown; p-values shown, one-way ANOVA Kruskal–Wallis test. (C) Ciliated respiratory cultures were infected with SARS-CoV-2 (MOI 0.05, $10^4$ PFU/Tissue) at 37°C and incubated for 24 h prior to continued incubation at 37 or temperature upshift to 40°C. Genome copies per ml of SARS-CoV-2 in apical washes collected at 72 h were determined RT-qPCR. $N = 9$ tissues per condition derived from a minimum of 3 independent biological experiments. Black line, median; whisker, 95% confidence interval; all data points shown; p-value shown, Mann–Whitney $U$ test. Raw values presented in S9 Data. HBEp, human bronchiolar epithelial; PFU, plaque-forming unit; RT-qPCR, reverse transcription quantitative PCR; SARS-CoV-2, Severe Acute Respiratory Syndrome Coronavirus 2.

observed at elevated temperature occurred due to viral genetic mutation. Amplicon sequencing of vRNA isolated from donor 1 apical washes at 72 h identified a total of 7 unique single-nucleotide polymorphisms (SNPs) relative to input sequence with no indels being identified (Table 1 and S2 Data). SNPs were only identified in genomic sequences isolated from apical washes incubated at 37°C (Table 1). Thus, the temperature-dependent restriction observed in SARS-CoV-2 replication at 40°C is not a consequence of viral mutation. As the generation of SNPs requires vRNA replication, the absence of genomic SNPs at 40°C may reflect a temperature-dependent block in vRNA replication. To investigate this, we quantified the intracellular levels of vRNA from within infected tissues. RT-qPCR analysis using 2 independent primer-probe sets (N and ORF1a) demonstrated significantly lower levels of intracellular vRNA in tissues incubated at 40°C relative to control samples (Fig 4C). ISH staining of tissue sections confirmed abundant levels of intraepithelial vRNA accumulation within all infected tissues incubated at 37°C, with little to no staining observed at 40°C (Fig 4D). Together, these data indicate that respiratory tissue remains permissive to SARS-CoV-2 infection (viral entry) but refractory to vRNA replication to levels sufficient for robust ISH detection.

## Elevated temperature restricts the onset of SARS-CoV-2 transcription independently of the IFN-mediated innate immune defenses

As we could detect significant levels of intracellular SARS-CoV-2 RNA within infected tissues at elevated temperature (Fig 4C), the temperature-dependent restriction observed could reflect enhanced levels of immune activation following pattern recognition receptor (PRR) detection of viral PAMPs [37,38]. To investigate this, we compared the relative transcriptome profiles from mock-treated or SARS-CoV-2–infected tissues at 37 or 40°C derived from 3 independent donors (Figs 5 and S1–S4, S1 and S3–S5 Data). Principle component analysis (PCA) demonstrated host transcriptome signatures to be donor specific (S1 Fig), with donor 3 (Caucasian female aged 16 years) demonstrating the most divergence from either of the other 2 donors

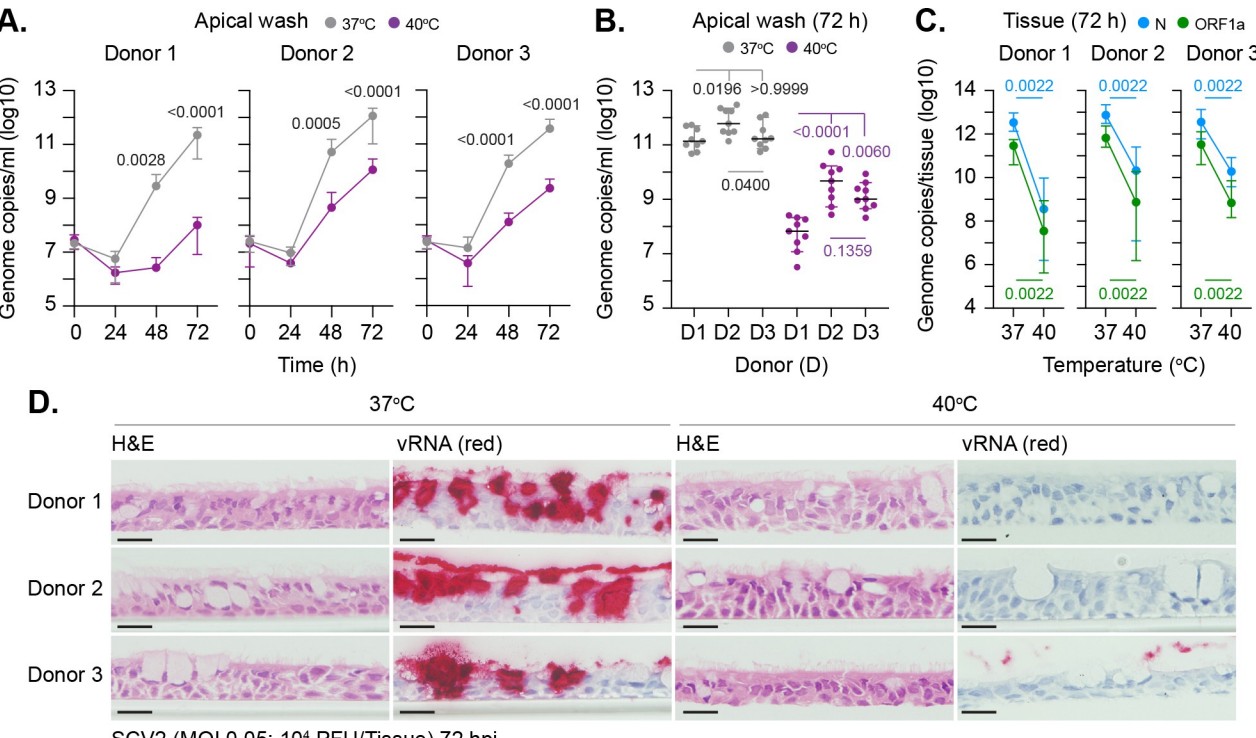

**Fig 4. Elevated temperature restricts SARS-CoV-2 replication in respiratory airway cultures in a donor-independent manner.** Ciliated respiratory cultures differentiated from primary HBEp cells isolated from 3 independent donors (donor 1, male Caucasian aged 63 years; donor 2, Hispanic male aged 62 years; donor 3, Caucasian female aged 16 years; all nonsmokers) were incubated at 37 or 40˚C for 24 h prior to SARS-CoV-2 (SCV2; MOI 0.05; $10^4$ PFU/Tissue) infection and incubation at the indicated temperatures. Apical washes were collected over time (as indicated) and tissues harvested at 72 h for RNA extraction or fixed for ISH staining. (A/B) Genome copies per ml of SARS-CoV-2 in apical washes harvested over time as determined RT-qPCR. $N = 9$ tissues per condition derived from a minimum of 3 independent biological experiments. (A) Means and SD shown; p-values shown, Mann–Whitney $U$ test between donor paired time points. (B) Black line, median; whisker, 95% confidence interval; all data points shown; p-values shown, one-way ANOVA Kruskal–Wallis test (top), Mann–Whitney $U$ test (bottom). (C) Quantitation of genome copies per tissue of SARS-CoV-2 at 72 h by RT-qPCR using 2 independent primer probe sets (N and ORF1a). $N = 6$ tissues per sample condition derived from a minimum of 3 independent biological experiments. Medians and 95% confidence interval shown; p-values shown, Mann–Whitney $U$ test. (D) Representative images of H&E and SARS-CoV-2 RNA ISH (red) stained sections. Hematoxylin was used as a counter stain. Scale bars = 20 μm. Raw values presented in S9 Data. HBEp, human bronchiolar epithelial; H&E, hematoxylin and eosin; ISH, in situ hybridization; PFU, plaque-forming unit; RT-qPCR, reverse transcription quantitative PCR; SARS-CoV-2, Severe Acute Respiratory Syndrome Coronavirus 2; vRNA, viral RNA.

(donor 1, Caucasian male aged 63 years; donor 2, Hispanic male aged 62 years). Analysis of pooled data identified distinct profiles of DEG (Q < 0.05) expression for each paired experimental condition analyzed (Fig 5A and 5B and S6 Data), with clusters of unique and shared DEG expression (Fig 5B, purple lines). Significant differences in lncRNA expression were also identified between each paired condition (S5 Fig and S7 Data). Pathway analysis identified distinct profiles of DEG enrichment, with infection at 37˚C inducing an elevated immune signature (immune system process, GO:0002376) relative to all other conditions (Fig 5C, arrow). Notably, SARS-CoV-2 infection at 40˚C failed to register a significant immune response above that observed for mock treatment (Fig 5C [arrow], gray box; p > 0.05). Collectively, these data identify tissue temperature to play an important role in the differential regulation of transcriptional host responses to SARS-CoV-2 infection, with enriched levels of immune gene expression observed during infection at 37˚C. Investigating further, we compared the transcriptome profiles from SARS-CoV-2–infected tissues at 37 and 40˚C. Out of the 1,934 DEGs (Q < 0.05) identified (Fig 6A), pathway analysis identified immune system and cytokine-related pathways to be suppressed during SARS-CoV-2 infection at 40 relative to 37˚C (Fig 6B and 6C and

**Table 1. SARS-CoV-2 sequence summary.** SARS-CoV-2 RNA isolated from apical washes collected from donor 1 infected tissues incubated at 37 or 40°C at 72 h were subjected to amplicon sequencing. Reference (ref) sequences (Wuhan, MN908947.3; England 2, EPI_ISL_407073) were used for consensus alignment and SNP identification. Unique SNPs (not detected in input and >5% base count frequency) and frequency (%base counts) shown. Full SNP analysis presented in S2 Data.

| Nucleotide position (Wuhan) | Base ref (Wuhan) | Base ref (England 2) | Base input (England 2) | SNP mutation vs. input (Sample ID) | SNP frequency (%) | Amino acid change (relative to Wuhan) |
|---|---|---|---|---|---|---|
| 11379 | C | C | C | C>T (37–2) | T (91.81) | nsp6 (A136V) |
| 15682 | T | T | T | T>C (37–2) | C (91.69) | nsp12 (Y748H) |
| 17725 | A | A | A | A>C (37–2) | C (91.85) | synonymous |
| 18488 | T | C | C | no change | C (>99.9) | nsp14 (I150T) |
| 23605 | T | G | G | no change | G (>99.9) | synonymous |
| 26558 | G | G | G | G>A (37–2) | A (90.74) | synonymous |
| 27631 | A | A | A | A>C (37–1) | C (73.49) | synonymous |
| 27892 | A | A | A | A>G (37–1) | G (73.99) | synonymous |
| 28114 | T | C | T | no change | T (>99.9) | no change |
| 29596 | A | G | G | no change | G (>99.9) | ORF10 (I13M) |
| 6945 | T | T | T | T>C (37–3) | C (71.77) | Nsp3 (I1409T) |
| 8782 | C | T | T | no change | T (>99.9) | synonymous |

SARS-CoV-2, Severe Acute Respiratory Syndrome Coronavirus 2; SNP, single-nucleotide polymorphism.

S8 Data). DEG analysis identified *IFNB1* (IFNβ) and *IFNL1-3* (IFNλ) transcript levels to only be up-regulated above 1 CPM during SARS-CoV-2 infection at 37°C (Fig 6D and S8 Data). While a small number of IFN-stimulated genes (ISGs) were identified to be up-regulated within infected tissues at 40°C relative to mock treatment (9 ISGs in total; S3 Fig), no net increase in immune DEG expression was observed between mock or SARS-CoV-2–infected tissues at 40°C relative to mock treatment at 37°C (Fig 6C). Together, these data suggest that SARS-CoV-2 infection of respiratory epithelia at 40°C fails to induce a robust IFN response. To corroborate this result, we performed RT-qPCR on RNA extracted from mock or SARS-CoV-2–infected tissues derived from multiple donors. The induction of *IFNB1* and *IFNL1* expression was readily observed in SARS-CoV-2–infected tissues at 37°C, with no significant change observed during infection at 40°C relative to mock-treated samples (Figs 6E and S6). Similarly, significant levels of ISG induction (*Mx1*, *ISG15*, and *IFIT2*) were only observed in SARS-CoV-2–infected tissues at 37°C (Figs 6F and S6). Notably, the levels of ISG induction at 37°C were relatively weak (3- to 6-fold relative to mock treatment; Fig 6F), indicative of low levels of innate immune stimulation at 72 h postinfection [20,39]. Infection of respiratory cultures treated with the Janus kinase (JAK) inhibitor Ruxolitinib (Ruxo; 5 μM), which blocks IFN-mediated JAK–STAT signaling during virus infection [40,41], demonstrated no significant change in SARS-CoV-2 apical titers relative to carrier control (DMSO) treatment (Fig 6G). Thus, in contrast to temperature elevation (Figs 3 and 4), these data demonstrate IFN-mediated innate immune defenses to have a negligible effect on viral yield up to 72 h postinfection in our model system. We next examined the influence of temperature on SARS-CoV-2 replication in VeroE6 cells, a *Chlorocebus sabaeus* (African green monkey) cell line known to be permissive to SARS-CoV-2 infection but defective in type-I IFN-mediated host defenses [26,42,43]. Temperature elevation to 40°C restricted SARS-CoV-2 replication, with or without preincubation at elevated temperature, relative to infection at 37°C (S7 Fig). Collectively, these data demonstrate the temperature-dependent restriction of SARS-CoV-2 replication to occur independently of IFN-mediated antiviral immune defenses.

We next investigated at which stage in the replication cycle SARS-CoV-2 became restricted at elevated temperature. RNA-Seq analysis of SARS-CoV-2–infected tissues demonstrated no

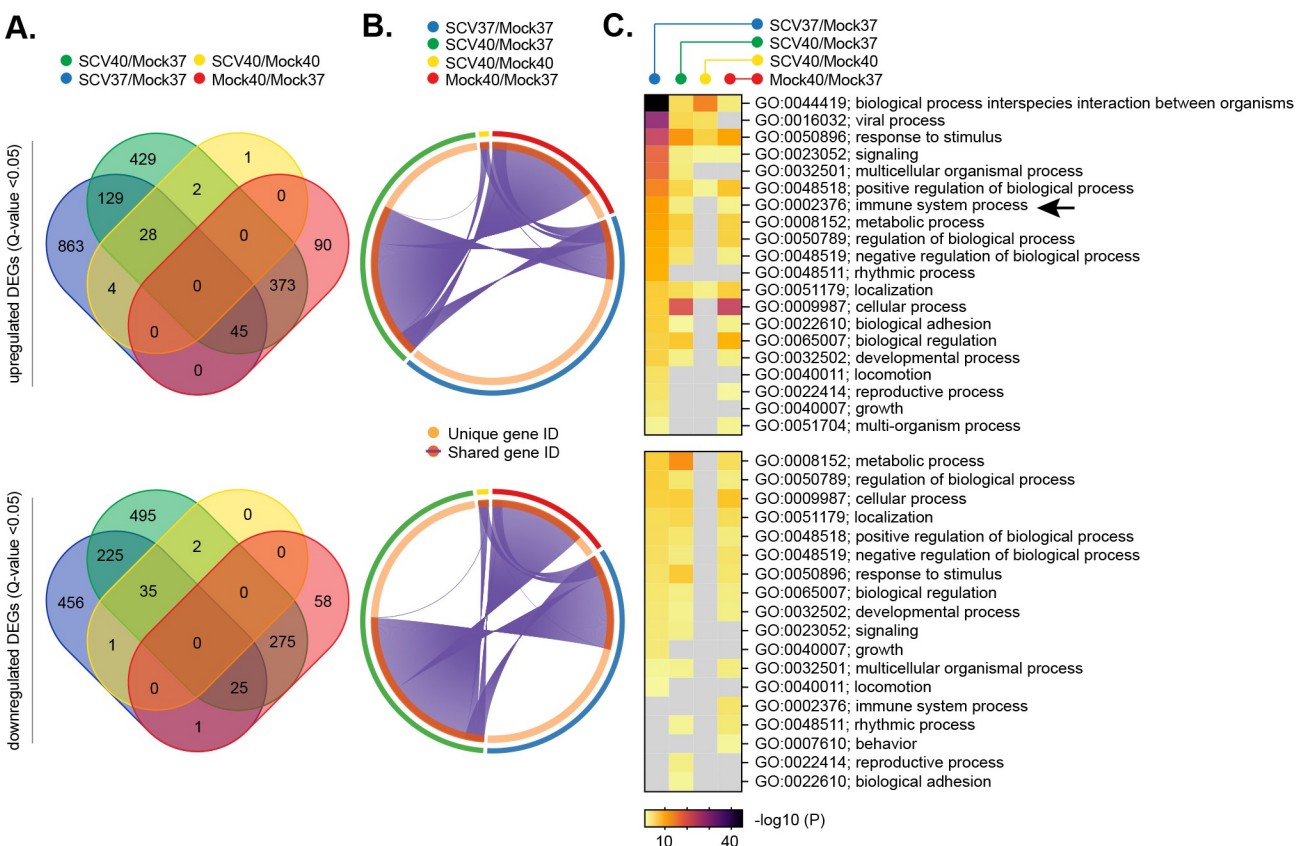

**Fig 5. Respiratory airway cultures induce distinct transcriptional host responses to SARS-CoV-2 infection at elevated temperature.** Ciliated respiratory cultures differentiated from primary HBEp cells isolated from 3 independent donors (donor 1, male Caucasian aged 63 years; donor 2, Hispanic male aged 62 years; donor 3, Caucasian female aged 16 years; all nonsmokers) were incubated at 37 or 40°C for 24 h prior to mock (media only) or SARS-CoV-2 (SCV2; MOI 0.05, $10^4$ PFU/Tissue) infection. Tissue were incubated at their respective temperatures for 72 h prior to RNA extraction and RNA-Seq. DEGs (Q < 0.05, up-regulated [top panels] or down-regulated [bottom panels]) were identified for each paired condition analyzed; blue ellipses, SARS-CoV-2 37°C/Mock 37°C (SCV37/Mock37); green ellipses, SARS-CoV-2 40°C/Mock 40°C (SCV40/Mock40); yellow ellipses, SARS-CoV-2 40°C/Mock 40°C (SCV40/Mock37); red ellipses, Mock 40°C/Mock 37°C (Mock40/Mock37). (A) Venn diagram showing the number of unique or shared DEGs between each paired condition analyzed. (B) Circos plot showing the proportion of unique (light orange inner circle) or shared (dark orange inner circle + purple lines) DEGs between each paired condition analyzed. (C) Metascape pathway analysis showing significant DEG enrichment *p*-value <0.05 (−log10 *p*-value shown) for each paired condition. Black arrow, immune system process pathway (GO:0002376). Gray boxes, *p* > 0.05. (A to C) RNA-Seq data derived from RNA isolated from 3 donors (donors 1 to 3) per sample condition derived from 3 independent experiments per donor. Raw values presented in S1, S3 to S6, and S9 Data. DEG, differentially expressed gene; HBEp, human bronchiolar epithelial; PFU, plaque-forming unit; SARS-CoV-2, Severe Acute Respiratory Syndrome Coronavirus 2.

difference in the total number of mapped reads (MRs; human + viral) between 37 and 40°C (Fig 7A, *p* = 0.5690; S9 Data). However, a significant decrease in the number of viral MRs spanning multiple SARS-CoV-2 ORFs was detected (Fig 7B–7D). Importantly, a substantial decrease in the expression of viral subgenomic RNAs (sgRNAs) was also detected at elevated temperature (Fig 7E, *p* = 0.0018). As the expression of SARS-CoV-2 sgRNAs requires the onset of viral transcription [44], these data identify tissue temperature to play an important role in the regulation of viral transcription. Correspondingly, indirect immunofluorescence staining of tissue sections demonstrated significantly fewer SARS-CoV-2 nucleocapsid (N) positive foci in tissues infected at 40 relative to 37°C, confirming reduced levels of viral gene expression (Fig 7F and 7G, *p* < 0.0001). Costaining of tissue sections for Mx1 expression, a well-established ISG product (Fig 6D and 6F), demonstrated Mx1 expression to be localized in proximity to SARS-CoV-2 infectious foci at 37°C, with little to no staining observed within infected

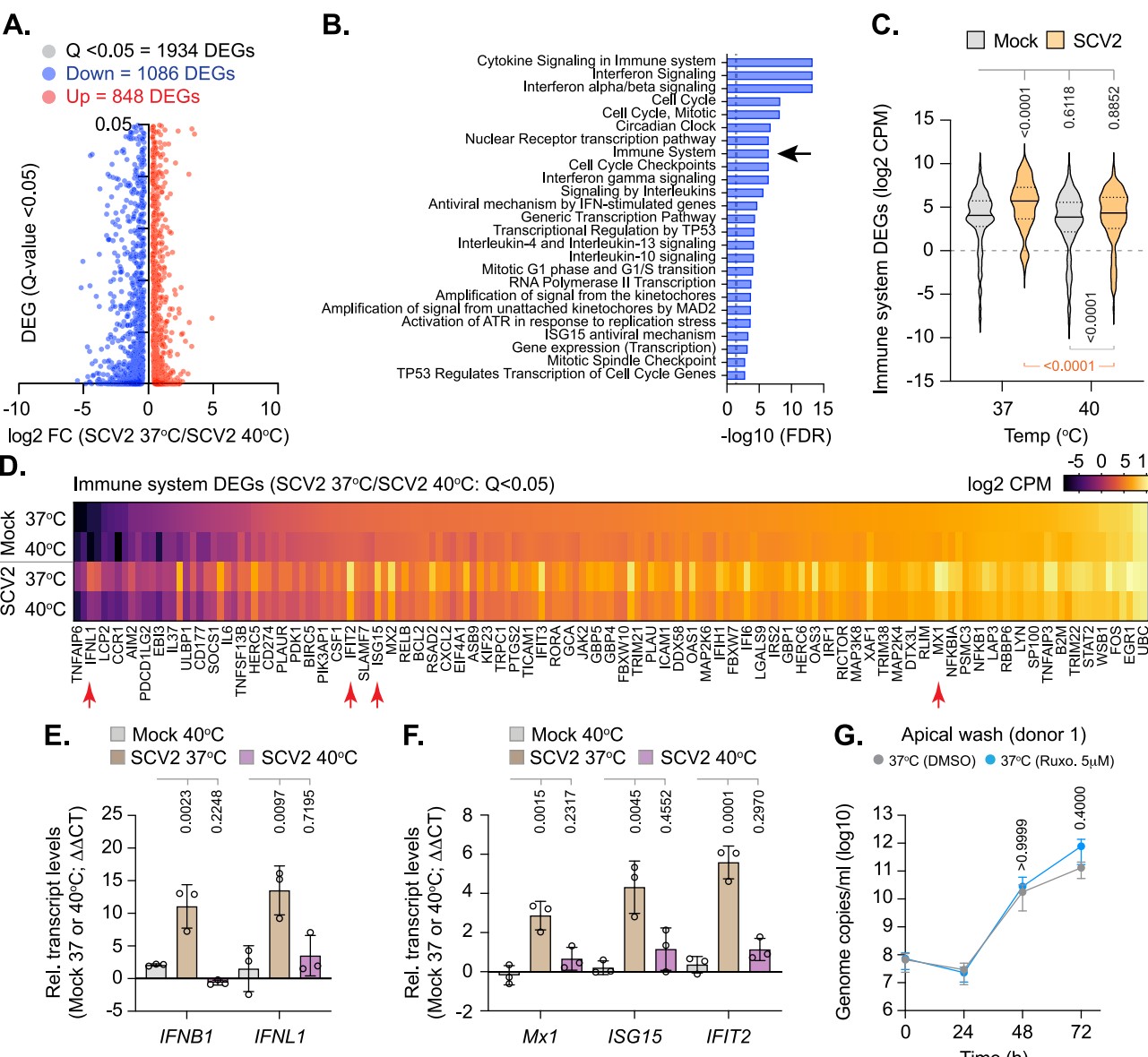

**Fig 6. Elevated temperature restricts SARS-CoV-2 replication in respiratory airway cultures independently of IFN-mediated antiviral immune defenses.** Ciliated respiratory cultures differentiated from primary HBEp cells isolated from 3 independent donors (donor 1, male Caucasian aged 63 years; donor 2, Hispanic male aged 62 years; donor 3, Caucasian female aged 16 years; all nonsmokers) were incubated at 37 or 40°C for 24 h prior to mock (media) or SARS-CoV-2 (SCV2; MOI 0.05, $10^4$ PFU/Tissue) infection. Tissues were incubated at their respective temperatures for 72 h prior to RNA extraction and RNA-Seq or RT-qPCR. (A) Scatter plots showing high confidence (Q < 0.05) DEG transcripts identified between SARS-CoV-2–infected respiratory cultures at 37 or 40°C; up-regulated DEGs, red circles; down-regulated DEGs, blue circles. (B) Reactome pathway analysis of mapped down-regulated DEGs. Top 25 down-regulated (FDR < 0.05) pathways shown (blue bars; plotted as −log10 FDR). Dotted line, threshold of significance (−log10 FDR of 0.05). (C) Expression profile (log2 CPM) of down-regulated immune system DEGs (R-HSA168256; arrow in B) relative to expression levels in mock tissue at 37 or 40°C. Black line, median; dotted lines, fifth and 95th percentile range; p-values shown, one-way ANOVA (top), paired two-tailed t test (bottom). (D) Expression levels (log2 CPM) of immune system DEGs (identified in B, black arrow) relative to mock at 37 or 40°C. Every second gene labeled. Red arrows highlight genes chosen for RT-qPCR validation. (A to D) RNA-Seq data derived from RNA isolated from 3 donors (donors 1 to 3) per sample condition derived from 3 independent experiments per donor. (E/F) RT-qPCR (ΔΔCT) quantitation of IFN (*IFNB1* and *IFNL1*) or ISG (*Mx1*, *ISG15*, and *IFIT2*) transcript levels within mock or SARS-CoV-2–infected respiratory cultures derived from 3 independent donors (transcript values normalized to GAPDH per sample condition). N ≥ 3 tissues per donor per condition derived from 3 independent biological experiments. Means and SD shown; all data points shown; p-values shown, one-way ANOVA. Individual ΔCT values per donor are shown in S6 Fig for donor comparison. (G) Respiratory cultures differentiated from donor 1 HBEp cells were pretreated with Ruxo (5 μM) or carrier control (DMSO) for 16 h prior to SARS-CoV-2 infection (MOI 0.05, $10^4$ PFU/Tissue) and continued incubation at 37°C in the presence of inhibitor or carrier control. Apical washes were collected at the indicated times (h) and genome copies per ml determined RT-qPCR. N = 3 tissues per sample condition derived from 3 independent biological experiments. Means and SD shown; p-values shown, Mann–Whitney U test. Raw values presented in S8 and S9 Data. CPM,

counts per million; DEG, differentially expressed gene; FDR, false discovery rate; HBEp, human bronchiolar epithelial; IFN, interferon; ISG, IFN-stimulated gene; PFU, plaque-forming unit; RT-qPCR, reverse transcription quantitative PCR; Ruxo, Ruxolitinib; SARS-CoV-2, Severe Acute Respiratory Syndrome Coronavirus 2.

respiratory epithelia at 40˚C (Fig 7F). We conclude that infection of respiratory tissue at elevated temperature restricts SARS-CoV-2 replication through a mechanism that inhibits viral transcription independently of IFN-mediated antiviral immune defenses previously shown to restrict SARS-CoV-2 replication [37,38].

## Discussion

A defining symptom of COVID-19 is the onset of fever with a febrile temperature range of 38 to 41˚C [4–8]. However, the effect of elevated temperature on SARS-CoV-2 tissue tropism and replication has remained to be determined. Here, we identify a temperature-sensitive

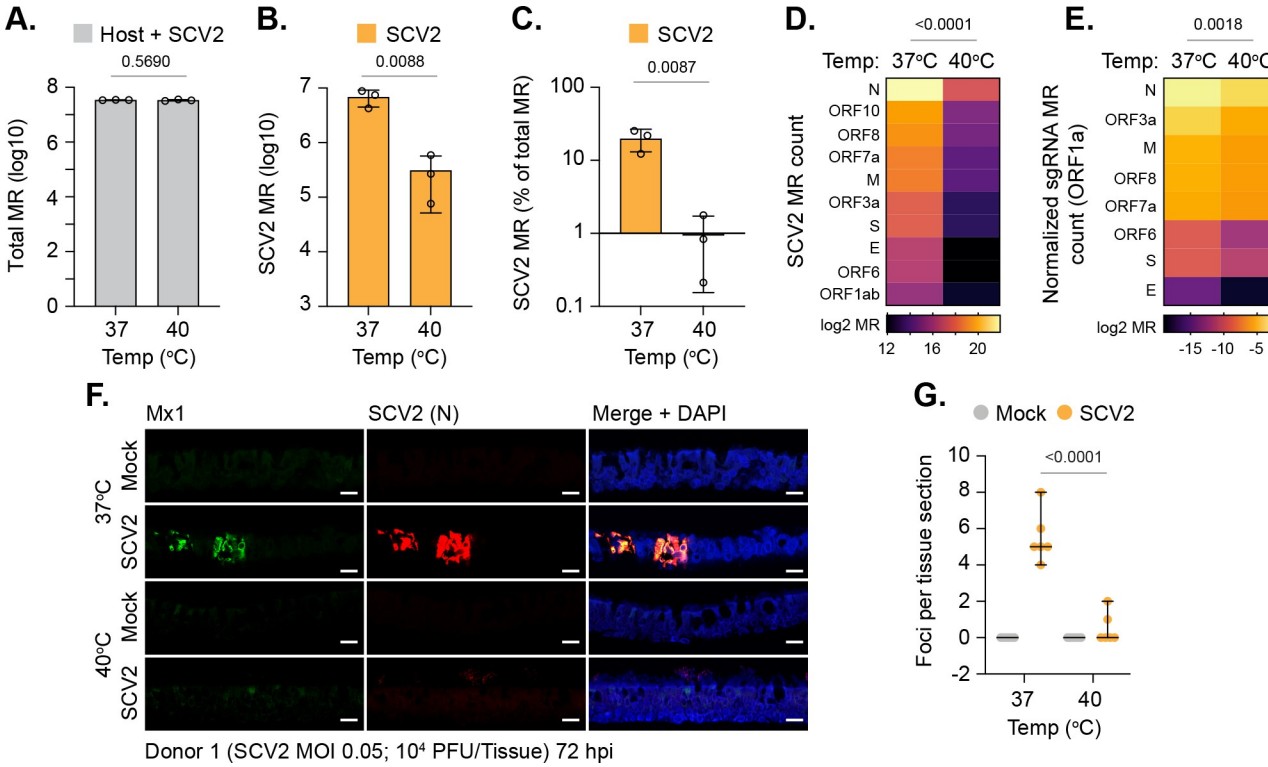

**Fig 7. Elevated temperature restricts SARS-CoV-2 transcription in respiratory airway cultures.** Ciliated respiratory cultures differentiated from primary HBEp cells isolated from 3 independent donors (donor 1, male Caucasian aged 63 years; donor 2, Hispanic male aged 62 years; donor 3, Caucasian female aged 16 years; all nonsmokers) were incubated at 37 or 40˚C for 24 h prior to mock (media only) or SARS-CoV-2 (SCV2; MOI 0.05, $10^4$ PFU/Tissue) infection. Tissues were incubated at their respective temperatures for 72 h prior to RNA extraction and RNA-Seq. (A) Total MRs (host + SARS-CoV-2) from infected tissues; means and SD shown; p-value shown, unpaired two-tailed t test. (B) SARS-CoV-2 MRs from infected tissues; means and SD shown. (C) % SARS-CoV-2 MRs of total MR count (human + SARS-CoV-2); means and SD shown. (B/C) Unpaired two-tailed t test, p-value shown. (D) Expression values (log2 MR) of SARS-CoV-2 ORFs. (E) Expression values (log2 sgRNA count normalized to ORF1a) of SARS-CoV-2 sgRNAs. (D/E) Paired two-tailed t test, p-values shown. (A to E) RNA-Seq data derived from RNA isolated from 3 donors (donors 1 to 3) per sample condition derived from 3 independent experiments per donor. (F) Indirect immunofluorescence staining of tissue sections (donor 1) showing ISG Mx1 (green) and SARS-CoV-2 nucleocapsid (N, red) expression. Nuclei were stained with DAPI. Scale bars = 20 μm. (G) Quantitation of SARS-CoV-2 N foci in respiratory tissue sections (as in F). N = 6 tissue sections per sample condition. Black line, median; whisker, 95% confidence interval; all data points shown; p-value shown, unpaired two-tailed t test. Raw values presented in S9 Data. HBEp, human bronchiolar epithelial; ISG, IFN-stimulated gene; MR, mapped read; PFU, plaque-forming unit; SARS-CoV-2, Severe Acute Respiratory Syndrome Coronavirus 2; sgRNA, subgenomic RNA.

phenotype in SARS-CoV-2 replication in respiratory epithelia that occurs independently of the robust induction of IFN-mediated innate immune defenses known to restrict SARS-CoV-2 replication [37,38]. The differentiation of stratified respiratory epithelium has proven to be a valuable research tool to investigate the cellular tropism, replication kinetics, and immune regulation of SARS-CoV-2, as these tissue models mimic many aspects of infection observed in animal models and COVID-19 patients [16–24,29,45–49]. While the use of such 3D models represents an important advancement over traditional two-dimensional cell culture systems, the absence of circulating immune cells (e.g., macrophages, natural killer cells, dendritic cells, and neutrophils), which modulate fever and proinflammatory immune responses to infection is an important limiting factor [9,10,50]. Thus, we limit our conclusions to the effect of elevated temperature on SARS-CoV-2 infection and replication within respiratory epithelium.

Consistent with previous reports [16,17,19–21,30,38,49], SARS-CoV-2 infection of respiratory epithelium at 37˚C induced a proinflammatory immune signature (S2D Fig; IFNβ and IFNλ1–3). While SARS-CoV-2 is known to induce a comparatively weak immune response relative to other respiratory viruses (e.g., IAV) [20,39], we observed localized areas of epithelial infection (Fig 1A, arrows) that were coincident with elevated levels of ISG expression by 72 h postinfection (Fig 7F, Mx1). These findings are consistent with reports that have shown localized areas of infection in the respiratory airway of infected ferrets and surface epithelium of organoid cultures [29–31,49]. Thus, the proportion of epithelial infection and/or rate of intraepithelial spread may account for the differences observed in immune signature reported between different respiratory pathogens [20,51]. As we observed localized areas of ISG induction, our data support a model of infection where cytokine (e.g., IFN) receptor-binding consumption kinetics may be a rate limiting factor in the protection of respiratory epithelia to SARS-CoV-2 infection [52,53]. Such observations warrant further investigation to determine whether these localized areas of epithelial immune protection are induced by specific cytokines (e.g., IFNβ or IFNλ1–3) and/or correlate with COVID-19 disease progression.

We demonstrate 3D respiratory epithelial cultures under ALI to induce a robust heat stress response upon temperature elevation without loss of epithelium integrity, disruption to general cellular transcription, or damage-associated molecular pattern (DAMP) activation of innate immune defenses (Figs 2 and S4). Temperature elevation alone was not sufficient to block SARS-CoV-2 entry (Figs 3, 4, and S7), but refractory to SARS-CoV-2 transcription leading to reduced levels of vRNA accumulation and apical shedding of infectious virus (Figs 3, 4, and 7). Thus, we identify febrile temperatures associated with COVID-19 to play an important role in limiting the epithelial replication of SARS-CoV-2 infection. Importantly, SARS-CoV-2 restriction occurred independently of the robust induction of type-I (IFNβ) or type-III (IFNλ) IFN-mediated immune defenses (Figs 5, 6, and S7), despite infected tissues having abundant levels of intracellular vRNA (Fig 4C). We posit that the lack of immune induction observed at 40˚C is likely a consequence of diminished levels of vRNA replication, which has been shown to play an important role in the production of PAMPs (e.g., dsRNA intermediates) required for the activation of innate immune defenses to other respiratory pathogens [54–57]. However, PRR detection of PAMPs plays an important role in the induction of a proinflammatory cytokines (including IL-6) required to mount a fever response to microbial challenge in vivo [9,10]. Thus, additional animal studies will be required to determine whether the temperature-dependent restriction of SARS-CoV-2 occurs independently of an IFN response in vivo. While speculative, we posit that low to moderate grade fever may confer protection to respiratory tissue within SARS-CoV-2–infected individuals as a component of a homeostatically controlled non-hyperinflammatory immune response to infection.

While we identify tissue temperature to play an important role in the regulation of SARS-CoV-2 transcription and replication in respiratory epithelia (Fig 7D and 7E), the precise

molecular mechanism(s) of restriction remain to be determined. The heat stress response has been shown to play both a positive (proviral) and negative (antiviral) role during virus replication [58–60]. Our transcriptomic analysis identified respiratory epithelial host responses to be differentially regulated in response to both temperature elevation and SARS-CoV-2 infection (Figs 5, 6, and S1–S5). Thus, SARS-CoV-2 infection of respiratory epithelia at elevated temperature elicits a distinct host response that may contribute to the restriction of SARS-CoV-2 replication at multiple stages of infection. For example, we identify lncRNAs to be differentially expressed in response to both SARS-CoV-2 infection and temperature (S5 Fig), which are known to influence the outcome of virus infection independently of IFN-mediated immune defenses [61]. Thus, multiple gene products and/or pathways may contribute to the sequential or accumulative restriction of SARS-CoV-2 replication within respiratory epithelia at elevated temperature.

We identify temperature elevation to correlate with lower levels of SARS-CoV-2 sgRNA expression (Fig 7E), identifying a temperature-dependent block in SARS-CoV-2 transcription. We hypothesize that this restriction may relate to an inhibition in SARS-CoV-2 RNA-dependent RNA polymerase (RdRp) activity and/or binding affinity to vRNA, as the genomic replication activity of the IAV RdRp polymerase is known to be restricted at elevated temperature (41°C) [62]. Notably, avian IAV cold adaptation of RdRp is a host determinant of IAV zoonosis, which naturally replicates in the intestinal tract of birds at higher temperatures [63]. As circulating strains of SARS-CoV-2 are susceptible to nonsynonymous mutation within RdRp coding sequences (http://cov-glue.cvr.gla.ac.uk/#/home), molecular studies are warranted to determine if such amino acid substitutions might influence the thermal restriction of SARS-CoV-2 replication. While SARS-CoV-2 has been reported to replicate to higher viral titers at temperatures associated with the upper respiratory airway (33°C) relative to core body temperature (37°C) [16], the replication of coronaviruses at febrile body temperatures has remained poorly characterized. While temperature-sensitive (ts) coronavirus mutants have been identified for feline infectious peritonitis virus (tsFIPV) and murine hepatitis virus (MHV tsNC11), their parental wild-type derivatives were able to replicate at temperatures ≥39°C [64,65]. The replication defect for tsFIPV was attributed to a block in viral maturation that limited its replication in cats to the upper respiratory airway [64], whereas the growth defect in tsNC11 was attributed to a coding mutation within the macrodomain and papain-like protease 2 domain of the nonstructural protein 3 [65]. Thus, we present the first evidence demonstrating a circulating strain of coronavirus to be sensitive to temperature thermoregulation. Genomic analysis of SARS-CoV-2 isolated from apical washes failed to identify any unique SNPs from infected tissues at 40°C relative to input sequence. These data support the temperature-dependent block in SARS-CoV-2 propagation to occur prior to vRNA replication, a prerequisite requirement for genomic mutation. As such, heat adaptation gain of function experiments through serial passage of SARS-CoV-2 at elevated temperature (≥39°C) may shed light on whether the restriction observed is related to viral and/or cellular host factors. Importantly, however, appropriate levels of biosafety (ethical and genetic modification) should be considered prior to such experimentation.

In summary, we identify an important role for tissue temperature in the cellular restriction of SARS-CoV-2 during infection of respiratory epithelia that occurs independently of the robust induction of canonical IFN-mediated antiviral immune defenses known to restrict SARS-CoV-2. We demonstrate tissue temperature to significantly influence the differential regulation of epithelial host responses to SARS-CoV-2 infection and the progression of viral transcription upon tissue infection. Future investigation is warranted to determine the precise mechanism(s) of restriction, as this may uncover novel avenues for therapeutic intervention in the treatment of COVID-19.

## Materials and methods

### Virus

SARS-CoV-2 (SCV2) strain BetaCoV/England/02/2020/EPI_ISL_407073 was isolated from a COVID-19 patient (Public Health England). The virus was passaged 3 times in Vero E6 cells and genotype sequence confirmed by Illumina sequencing. All experiments were performed in a Biosafety level 3-laboratory at the MRC-University of Glasgow Centre for Virus Research (SAPO/223/2017/1a).

### SARS-CoV-2 replication kinetics in Vero E6 cells

Vero E6 cells (a gift from Michelle Bouloy, Institute Pasteur, France) were propagated in Dulbecco's Modified Eagle Medium (DMEM GlutaMAX; ThermoFisher, 31966–021) supplemented with 10% fetal calf serum (FCS; ThermoFisher, 10499044 [Lot Number 08G8293K]) at 37˚C with 5% $CO_2$. Cells were seeded the day before infection at a density of $2 \times 10^5$ cells per well in a 12-well plate and infected with $10^4$ PFU of SARS-CoV-2 diluted in serum-free DMEM for 120 min with occasional rocking. The inoculum was removed and replaced by DMEM + 10% FCS. Supernatant was collected every 24 h and snap frozen prior to analysis. For temperature inhibition experiments, cells were either preincubated at 37 or 40˚C for 24 h prior to infection with continued incubation at their respective temperatures or incubated at the indicated temperatures after infection at 37˚C after the addition of overlay medium (as indicated).

### Differentiation and infection of respiratory epithelium

Three independent donors of primary HBEp cells were used for this study. Donor 1, 63-year-old Caucasian male (EpithelixSarl, 02AB077201F2); donor 2, 62-year-old Hispanic male (EpithelixSarl, AB079301); donor 3, 16-year-old Caucasian female (Lonza, CC-2540S). All donors were healthy and nonsmokers. For expansion and amplification, cells were propagated at 37˚C with 5% $CO_2$ in hAEC culture medium (Epithelix, EP09AM). All differentiation experiments were performed on cells that had been passaged a total of 4 times. Cells were seeded onto transwells (Falcon, 734–0036) at a cell density of $3 \times 10^4$ cells per transwell and grown to confluency. Cells were differentiated under ALI in PneumaCult-ALI medium containing hydrocortisone and heparin supplements (STEMCELL Technologies, 05001) as specified by the manufacturer's guidelines. Cells were differentiated for a minimum of 35 days, with media changed every 48 h and washed twice weekly in serum-free DMEM after day 20 post-ALI to remove mucus build up. ALI cultures were apically washed twice before infection, 24 h prior to infection and immediately preceding infection. Cultures were inoculated apically with 100 μl of serum-free DMEM containing $10^4$ PFU (MOI of 0.05) of SARS-CoV-2 (based on Vero E6 titres) for 120 min. The inoculum was removed, and the apical surface washed once in serum-free DMEM (0 h time point). Unless stated otherwise, ALI cultures were incubated at 37˚C with 5% $CO_2$. For temperature inhibition experiments, ALI cultures were preincubated at 37 or 40˚C for 24 h prior to infection with continued incubation at their respective temperatures. For temperature upshift experiments, ALI cultures were infected at 37˚C and incubated for 24 h prior to temperature upshift to 40˚C or continued incubation at 37˚C. For Ruxolitinib (Ruxo) experiments, DMSO (carrier control) or Ruxo (Selleckchem, S1378) was added to basal medium at a final concentration of 5 μM for 16 h prior to infection at 37˚C. Cultures were incubated at 37˚C and basal media changed daily containing fresh inhibitor. Apical washes were collected every 24 h in 200 μl of serum-free DMEM incubated for 30 min to collect infectious virus. Samples were stored at −80˚C until required. Tissues were fixed at

the indicated time points in 8% formaldehyde (Fisher Scientific, F/1501/PB17) for 16 to 24 h prior to paraffin embedding and processing. TEER measurements were conducted using an ERS-2 Voltohmmeter (Millicell; MERS00002). PBS was added to apical and basal chambers of ALI cultures or empty transwells and allowed to stand at room temperature (RT) for 15 min. Readings were taken prior to temperature shift (day 0) and every 24 h for 3 days. TEER measurements were calculated by subtracting the average of empty transwells from ALI epithelial readings and reported as $\Omega cm^2$.

## $TCID_{50}$

Vero E6 cells (subclone 6F5, MESO) were seeded the day before infection into 96-well plates at a density of $1.1 \times 10^4$ cells/well in DMEM + 10% FCS at 37˚C with 5% $CO_2$. Virus supernatants were serially diluted 1:10 in DMEM + 2% FCS in a total volume of 100 μl. Cells were infected in triplicate and incubated for 3 days at 37˚C with 5% $CO_2$. Plates were incubated for 3 days at 37˚C in 5% $CO_2$. Cells were fixed in 8% formaldehyde (Fisher Scientific, F/1501/PB17) for ≥2.5 h. Cytopathic effect was scored by staining with 0.1% Coomassie Brilliant Blue (BioRad, 1610406) in 45% methanol and 10% glacial acetic acid. $TCID_{50}$ was calculated according to Reed–Muench and Spearman–Kärber as described [66].

## Plaque assay

Vero E6 cells (subclone 6F5, MESO) were seeded at a density of $2.5 \times 10^5$ cells/well in 12-well plates the day before infection. Ten-fold serial dilutions of virus were prepared in DMEM + 2% FCS in a final volume of 500 μl. Cells were infected for 60 min with occasional rocking at 37˚C in 5% $CO_2$. The inoculum was removed and 1 ml of overlay containing 0.6% Avicel (Avicel microcrystalline cellulose, RC-591) made up in DMEM containing 2% FCS was added to each well. Cells were incubated for 3 days at 37˚C in 5% $CO_2$ prior to fixation in 8% formaldehyde and Coomassie brilliant blue staining (as described above). Plaques were counted manually and the PFU per ml calculated.

## Reverse transcription quantitative PCR (RT-qPCR)

RNA was extracted from TRIzol treated apical wash samples using a QIAGEN RNeasy Mini kit (Qiagen, 74106) or in a 96-well plate format using the KingFisher Flex purification system (Thermo Scientific, 5400610) and RNAdvance kit (Beckman Coulter, A35604) following manufacturer's protocols. SARS-CoV-2 RNA was quantified using a NEB Luna Universal Probe One-Step RT-qPCR Kit (New England Biolabs, E3006) and 2019-nCoV CDC N1 primers and probes (IDT, 10006713) or against the Orf1a open reading frame (Thermofisher, 4448508. Forward 5′- GACATAGAAGTTACTGGCGATAG-3′. Probe: VIC- ACCCCGTGACCT TGGTGCTTGT-MGB/NFQ. Reverse: 5′- TTAATATGACGCGCACTACAG-3′). Genome copy numbers were quantified using a standard curve generated from serial dilutions of an RNA standard. The RNA standard was calibrated using a plasmid (2019-nCoV_N; IDT, 10006625) that was quantified using droplet digital PCR. Values were normalized to copies/ml for apical washes or copies/tissue for cell-associated RNA.

## IFN-β/λ and ISG expression analysis

For IFN-β and IFN-λ quantification, cDNA was prepared from extracted RNA using Maxima First Strand cDNA Synthesis kit for RT-qPCR (Thermofisher, K1672).

qPCR reactions were performed using TakyonTM Low Rox Probe MasterMix dTTP Blue (Eurogentec, B0701) with primers and probe detecting GAPDH (Forward: 5′- GAAGGTGAA

GGTCGGAGT -3′, Reverse:5′- GAAGATGGTGATGGGATTTC-3′, Probe: FAM- CAAG CTTCCCGTTCTCAGCC -TAMRA). IFN-β and IFN-λ were analyzed with Applied Biosystems, assay identification numbers Hs01077958_s1 and Hs00601677_g1, following manufacturer's instructions. For ISG analysis, cDNA was generated from extracted RNA using a high-capacity cDNA reverse transcription kit (Thermofisher, 4368814). qPCR reactions were performed using a Brilliant III Ultra-fast qPCR master mix (Agilent: 600884) to quantify GAPDH, Mx1, ISG15, and IFIT2/ISG54 expression using TaqMan gene-specific primer (FAM/MGB) probe mixes (Life Technologies): assay ID Mx1 (HS00895608_m1), ISG15 (Hs01921425_s1), IFIT2 (Hs01922738_s1), and GAPDH (4333764F).

## In situ hybridization and immunostaining of respiratory tissue sections

Paraffin-embedded tissues were sectioned using a microtome (approximately 2 to 3 μm thick) and mounted on glass slides. Tissue sections were stained with H&E. For immunofluorescence staining, tissue sections underwent antigen retrieval by citrate (pH 6) or EDTA (pH 8) pressure cooking (as indicated below). RNAscope was used for the detection of SARS-CoV-2 RNA using a Spike-specific probe set (Advanced Cell Diagnostics, 848561 and 322372) following the manufacturer's protocol, which included a pretreatment with boiling in target solution and proteinase K treatment. ACE2 was detected using a rabbit polyclonal anti-hACE2 antibody (Cell Signaling, 4355S; citrate antigen retrieval) and EnVision+ anti-rabbit HRP (Agilent, K4003). Slides were scanned with a bright field slide scanner (Leica, Aperio Versa 8). Mx1 was detected using a mouse monoclonal anti-Mx1 antibody ([67]; EDTA antigen retrieval). Nucleocapsid protein was detected using a sheep polyclonal anti-SARS-CoV-2 N protein antibody (University of Dundee, DA114; EDTA antigen retrieval). Secondary antibodies for detection were rabbit anti-sheep AlexaFluor 555 (abcam, ab150182) and rabbit anti-mouse AlexaFluor 488 (Sigma-Aldrich, SAB4600056). Nuclei were stained with ProLong Gold (Life Technologies, P36941). Images were collected using a Zeiss LSM 710 confocal microscope using 40× Plan-Apochromat oil objective lens (numerical aperture 1.4) using 405, 488, and 543 nm laser lines. Zen black software (Zeiss) was used for image capture and exporting images, with minimal adjustment (image rotation) in Adobe Photoshop for presentation.

## RNA sequencing (RNA-Seq)

Infected or mock-treated respiratory cultures derived from a minimum of 3 independent experiments per donor per sample condition were harvested by scraping into TRIzol Reagent and transferred into tubes containing 2.8 mm ceramic beads (Stretton Scientific, P000916-LYSK0-A.0). Samples were homogenized (two 20 s pulses with a 30-s interval at RT) using a Percellys Cryolys Evolution Super Homogenizer (Bertin Instruments, P000671-CLYS2-A) at 5,500 rounds per min. The homogenized suspension was loaded into QIAshredder tubes (Qiagen, 79654) and centrifuged (12,000 × $g$ for 2 min at RT). A volume of 0.25 ml of chloroform (VWR Life Sciences, 0757) was added to the eluate and incubated at RT for 7 min prior to centrifugation (12,000 × $g$, 15 min, 4˚C). The aqueous phase was transferred into a fresh tube, mixed with 250 μl of 100% ethanol, and RNA isolated using RNAeasy columns (Qiagen, 74104) following the manufacturer's protocol, which included a 15-min DNase treatment (Qiagen, 79254) treatment at RT. Eluted RNA was quantified using Qubit Fluorometer 4 (Life Technologies, Q33238), Qubit RNA HS Assay (Life Technologies, Q32855) and dsDNA HS Assay Kits (Life Technologies, Q32854) and quality controlled on a 4200 TapeStation System (Agilent Technologies, G2991A) with a High Sensitivity RNA Screen Tape assay (Agilent Technologies, 5067–5579). All samples had a RIN score of ≥8.8. Total RNA (220ng) was used to prepare libraries for sequencing using an Illumina TruSeq Stranded mRNA Library Prep kit

(Illumina, 20020594) and SuperScript II Reverse Transcriptase (Invitrogen, 18064014) according to the manufacturer's instructions. Dual indexed libraries were PCR amplified, then cleaned up with Agencourt AMPure XP magnetic beads (Beckman Coulter), quantified using Qubit Fluorometer 4 (Life Technologies, Q33238) and Qubit dsDNA HS Assay Kit (Life Technologies Q32854), and the size distribution assessed using a 4200 TapeStation System (Agilent Technologies, G2991A) with a High Sensitivity D1000 Screen Tape assay (Agilent Technologies, 5067–5584). Libraries were pooled in equimolar concentrations and sequenced using an Illumina NextSeq 500/550 sequencer (Illumina, FC-404-2005). At least 95% of the reads generated presented a Q score of ≥30. RNA-Seq reads were quality assessed using FastQC (http://www.bioinformatics.babraham.ac.uk/projects/fastqc)). Sequence adaptors removed using TrimGalore (https://www.bioinformatics.babraham.ac.uk/projects/trim_galore/)). RNA-Seq reads were aligned to the *Homo sapiens* genome (GRCh38), downloaded via Ensembl using HISAT2. HISAT2 is a fast and sensitive splice aware mapper, which aligns RNA-Seq reads to mammalian-sized genomes using FM index strategy [68]. FeatureCounts [69] was used to calculate MR counts that were normalized to counts per million (CPM; unless otherwise stated). Generalized linear models (GLMs) with multifactor designs was used for DEG analysis in EdgeR [70]. A false discovery rate (FDR) $p$-value <0.05 (Q < 0.05) was used as a cutoff of significant differential gene expression. Sequencing reads were aligned to SARS-CoV-2 (GenBank accession MN908947.3) using BWA [71]. Periscope was used to quantify sgRNA expression levels [72]. Values were normalized to ORF1a. Only high-confidence (Q < 0.05) DEGs were used for pathway analysis in Reactome (https://reactome.org) [73,74] or differential pathway enrichment in Metascape (https://metascape.org/gp/index.html#/main/step1) [75]. In Reactome, the gene mapping tool was used as a filter to identify pathways enriched (overrepresented) for mapped DEG entities. FDR values <0.05 were considered significant for pathway enrichment. In Metascape, all DEGs were used for differential pathway analysis. Pathway $p$-values <0.05 were considered significant. Differentially expressed lncRNAs (Q < 0.05) were identified using the EnsemblBioMarttool (http://www.ensembl.org/biomart/martview/05285d5f063a05a82b8ba71fe18a0f18). Heat maps were plotted in GraphPad Prism (version 9.2.0). Mean CPM values of 0 were normalized to 0.01 for log2 presentation. Venn diagrams were plotted using http://bioinformatics.psb.ugent.be/webtools/Venn/.

## Viral genomic amplicon sequencing

vRNA was extracted from input or extracellular released virus collected from apical washes (72 h postinfection) using TRIzol Reagent. RNA quality control was performed using Qubit (Invitrogen) and Agilent 4200 Tapestation System (Agilent). For amplicon-based sequencing, amplicons were prepared according to the protocol developed by the ARTIC Network using primer version 3 (https://artic.network/ncov-2019). DNA fragments were cleaned using AMPure beads (Beckman Coulter). Illumina sequencing libraries were prepared using 40 ng of amplicon and the KAPA HyperPrep kit (Roche, KK8502), as per manufacturer's protocol. Samples were indexed using the NEBNext Multiplex Oligos for Illumina Dual Index Primers Set 1 (New England Biolabs, E6440S) and libraries amplified with 8 cycles of PCR. Amplified libraries were cleaned using AMPure beads (Beckman Coulter) and quantified using Qubit dsDNA HS kit (Invitrogen) and run on an Agilent 4200 Tapestation System (Agilent). Equimolar amounts of libraries were pooled, loaded at a final concentration of 11 pM, and sequenced in an Illumina MiSeq System (Illumina) using a MiSeq v2 cartridge 500 cycle kit. Sequences have been deposited in the European Nucleotide Archive (https://www.ebi.ac.uk/ena/browser/home), accession number PRJEB41332. Reads were aligned against Wuhan-Hu-1 reference strain (GenBank accession MN908947.3) using BWA [71] followed by primer

trimming and consensus calling with iVar [76]. The assembled data were parsed using Diversi-Tools (http://josephhughes.github.io/btctools/) to determine the frequency of nucleotides at each site and observe the mutations. COV-GLUE (http://cov-glue.cvr.gla.ac.uk/#/home) was used for identifying amino acid replacement following by mutation.

## Statistical analysis

The number (N) of tissues sampled per experimental condition is shown throughout and derived from a minimum of 3 independent biological experiments. GraphPad Prism (version 9.2.0) was used for statistical analysis on all data points per sample condition. Statistical tests and $p$-values are shown throughout. Statistically significant differences were accepted at $p < 0.05$.

## Supporting information

**S1 Fig. PCA of SARS-CoV-2–infected respiratory cultures derived from 3 independent donors.** Ciliated respiratory cultures differentiated from primary HBEp cells isolated from 3 independent donors (donor 1, male Caucasian aged 63 years; donor 2, Hispanic male aged 62 years; donor 3, Caucasian female aged 16 years; all nonsmokers) were incubated at 37 or 40°C for 24 h prior to mock (media only) or SARS-CoV-2 (SCV2; MOI 0.05, $10^4$ PFU/Tissue) infection. Tissues were incubated at their respective temperatures for 72 h prior to RNA extraction and RNA-Seq. (A) PCA analysis of mean CPM values derived from host transcriptome mapping of donor samples (dotted circles; donors 1 to 3) per experimental condition (as indicated). (B) Total MR (host + SARS-CoV-2) counts of donor (D1 to D3) infected samples at 37 or 40°C (as indicated); $p$-values shown, one-way ANOVA. (C) SARS-CoV-2 MR counts of donor (D1 to D3) infected samples at 37 or 40°C; $p$-values shown; top, unpaired two-tailed $t$ test; bottom, one-way ANOVA. (A to C) RNA-Seq data derived from RNA isolated from 3 independent biological experiments per donor. Raw values presented in S9 Data. CPM, counts per million; HBEp, human bronchiolar epithelial; MR, mapped read; PCA, principle component analysis; PFU, plaque-forming unit; RNA-Seq, RNA sequencing; SARS-CoV-2, Severe Acute Respiratory Syndrome Coronavirus 2.
(TIF)

**S2 Fig. SARS-CoV-2 infection of respiratory airway cultures induces a type-I and type-III IFN response.** Ciliated respiratory cultures differentiated from primary HBEp cells isolated from 3 independent donors (donor 1, male Caucasian aged 63 years; donor 2, Hispanic male aged 62 years; donor 3, Caucasian female aged 16 years; all nonsmokers) were mock (media only) or SARS-CoV-2 (SCV2; MOI 0.05, $10^4$ PFU/Tissue) infected at 37°C for 72 h prior to RNA extraction and RNA-Seq. (A) Scatter plots showing high-confidence (Q < 0.05) DEG transcripts identified between mock and SARS-CoV-2–infected cultures; up-regulated DEGs, red circles; down-regulated DEGs, blue circles. (B) Reactome pathway analysis of mapped up-regulated DEGs. Top 30 up-regulated (FDR < 0.05) pathways shown (red bars; plotted as −log10 FDR). Dotted line, threshold of significance (−log10 FDR of 0.05). (C) Expression profile (log2 CPM) of immune system–related DEGs (R-HSA168256; arrow in B). Black line, median; dotted lines; fifth and 95th percentile range; $p$-value shown, paired two-tailed $t$ test. (D) Expression levels (log2 CPM) of individual immune system DEGs (arrow in B). Every second gene labeled. (A to D) RNA-Seq data derived from RNA isolated from 3 donors (donors 1 to 3) per sample condition from 3 independent biological experiments per donor. Raw values presented in S3 and S9 Data. CPM, counts per million; DEG, differentially expressed gene; FDR, false discovery rate; HBEp, human bronchiolar epithelial; IFN, interferon; PFU, plaque-

forming unit; RNA-Seq, RNA sequencing; SARS-CoV-2, Severe Acute Respiratory Syndrome Coronavirus 2.
(TIF)

**S3 Fig. Identification of DEGs in mock and SARS-CoV-2–infected respiratory cultures incubated at 40˚C.** Ciliated respiratory cultures differentiated from primary HBEp cells isolated from 3 independent donors (donor 1, male Caucasian aged 63 years; donor 2, Hispanic male aged 62 years; donor 3, Caucasian female aged 16 years; all nonsmokers) were incubated at 40˚C for 24 h prior to mock (media only) or SARS-CoV-2 (SCV2; MOI 0.05, $10^4$ PFU/Tissue) infection and continued incubation at 40˚C. Tissues were harvested at 72 h for RNA extraction and RNA-Seq. (A) Scatter plots showing high-confidence (Q < 0.05) DEG transcripts identified between mock and SARS-CoV-2–infected respiratory cultures at 40˚C; up-regulated DEGs, red circles; down-regulated DEGs, blue circles. (B) Reactome pathway analysis of mapped up-regulated DEGs. Up-regulated pathways shown (red bars; plotted as −log10 FDR). Dotted line, threshold of significance (−log10 FDR of 0.05). (C) Expression values (log2 CPM) of Reactome mapped DEGs (identified in A); *p*-values shown, paired two-tailed *t* test. (A to C) RNA-Seq data derived from RNA isolated from 3 donors (donors 1 to 3) per sample condition from 3 independent biological experiments per donor. Raw values presented in S4 and S9 Data. CPM, counts per million; DEG, differentially expressed gene; FDR, false discovery rate; HBEp, human bronchiolar epithelial; IFN, interferon; ISG, IFN-stimulated gene; PFU, plaque-forming unit; RNA-Seq, RNA sequencing; SARS-CoV-2, Severe Acute Respiratory Syndrome Coronavirus 2.
(TIF)

**S4 Fig. Identification of DEGs in mock and SARS-CoV-2–infected respiratory cultures incubated at 37 and 40˚C, respectively.** Ciliated respiratory cultures differentiated from primary HBEp cells isolated from 3 independent donors (donor 1, male Caucasian aged 63 years; donor 2, Hispanic male aged 62 years; donor 3, Caucasian female aged 16 years; all nonsmokers) were incubated at 37 or 40˚C for 24 h prior to mock (media only) or SARS-CoV-2 (SCV2; MOI 0.05, $10^4$ PFU/Tissue) infection and continued incubation at their respective temperatures (as indicated). Tissues were harvested at 72 h for RNA extraction and RNA-Seq. (A) Scatter plots showing high-confidence (Q < 0.05) DEG transcripts identified between mock (37˚C) and SARS-CoV-2–infected (40˚C) respiratory cultures; up-regulated DEGs, red circles; down-regulated DEGs, blue circles. (B) Reactome pathway analysis of mapped DEGs. Top 15 up-regulated and down-regulated (FDR < 0.05; red and blue bars, respectively; plotted as −log10 FDR) pathways shown. Dotted line, threshold of significance (−log10 FDR of 0.05). Black arrow, identification of cellular response to heat stress pathway. (C) Expression values (log2 CPM) of Reactome mapped DEGs; *p*-values shown, paired two-tailed *t* test. Every 30th gene labeled. (A to C) RNA-Seq data derived from RNA isolated from 3 donors (donors 1 to 3) per sample condition from 3 independent biological experiments per donor. Raw values presented in S5 and S9 Data. CPM, counts per million; DEG, differentially expressed gene; FDR, false discovery rate; HBEp, human bronchiolar epithelial; PFU, plaque-forming unit; RNA-Seq, RNA sequencing; SARS-CoV-2, Severe Acute Respiratory Syndrome Coronavirus 2.
(TIF)

**S5 Fig. Respiratory airway cultures induce distinct lncRNA and miRNA transcriptional host responses to SARS-CoV-2 infection at elevated temperature.** Ciliated respiratory cultures differentiated from primary HBEp cells isolated from 3 independent donors (donor 1, male Caucasian aged 63 years; donor 2, Hispanic male aged 62 years; donor 3, Caucasian female aged 16 years; all nonsmokers) were incubated at 37 or 40˚C for 24 h prior to mock

(media only) or SARS-CoV-2 (SCV2; MOI 0.05, $10^4$ PFU/Tissue) infection. Tissues were incubated at their respective temperatures for 72 h prior to RNA extraction and RNA-Seq. High-confidence (Q < 0.05) DEG transcripts were identified (up-regulated [top panels] or down-regulated [bottom panels]) for each paired condition analyzed; blue circles/ellipses, SARS-CoV-2 37˚C/Mock 37˚C (SCV37/Mock37); green circles/ellipses, SARS-CoV-2 40˚C/Mock 40˚C (SCV40/Mock40); red circles/ellipses, SARS-CoV-2 40˚C/Mock 37˚C (SCV40/Mock37); yellow circles/ellipses, Mock 40˚C/Mock 37˚C (Mock40/Mock37). (A) Proportion of lncRNA (gray numbers and lines) DEGs identified per condition analyzed (colored numbers and lines). (B) Venn diagram showing the number of shared lncRNA identified between each paired condition analyzed. (C) Circos plot showing the proportion of unique (light orange inner circle) or shared (dark orange inner circle + purple lines) lncRNA between each paired condition analyzed. (D) Expression values (log2 CPM) of lncRNA identified per sample condition analyzed; *p*-values shown, paired two-tailed *t* test. (A to D) RNA-Seq data derived from RNA isolated from 3 donors (donors 1 to 3) per sample condition from 3 independent biological experiments per donor. Raw values presented in S7 and S9 Data. CPM, counts per million; DEG, differentially expressed gene; HBEp, human bronchiolar epithelial; lncRNA, long noncoding RNA; miRNA, microRNA; PFU, plaque-forming unit; RNA-Seq, RNA sequencing; SARS-CoV-2, Severe Acute Respiratory Syndrome Coronavirus 2.
(TIF)

**S6 Fig. Elevated temperature restricts SARS-CoV-2 replication in respiratory airway cultures in a donor- and IFN-independent manner.** Ciliated respiratory cultures differentiated from primary HBEp cells isolated from 3 independent donors (donor 1 [D1], male Caucasian aged 63 years; donor 2 [D2], Hispanic male aged 62 years; donor 3 [D3], Caucasian female aged 16 years) were incubated at 37 or 40˚C for 24 h prior to mock (media only) or SARS-CoV-2 (SCV2; MOI 0.05, $10^4$ PFU/Tissue) infection. Tissues were incubated at their respective temperatures for 72 h prior to RNA extraction and RT-qPCR. (A/B) Quantitation of IFN (*IFNB1* and *IFNL1*) or ISG (*Mx1*, *ISG15*, and *IFIT2*) transcript levels within mock of SARS-CoV-2–infected respiratory cultures. $N \geq 3$ tissues per sample condition derived from a minimum of 3 independent biological experiments. Means and SD shown; all data points (normalized [GAPDH] ΔCT values) shown; *p*-values shown, one-way ANOVA Kruskal–Wallis test. Dotted line, threshold of linear assay detection. ΔCT values presented to aid comparison between donor samples relative to that of averaged ΔΔCT values presented in Fig 6E and 6F per experimental condition. Raw values presented in S9 Data. HBEp, human bronchiolar epithelial; IFN, interferon; ISG, IFN-stimulated gene; PFU, plaque-forming unit; RT-qPCR, reverse transcription quantitative PCR; SARS-CoV-2, Severe Acute Respiratory Syndrome Coronavirus 2.
(TIF)

**S7 Fig. Elevated temperature restricts SARS-CoV-2 replication in Vero E6 cells.** Vero E6 cells were infected with SARS-CoV-2 (SCV2; MOI 0.05 PFU/cell) at 37˚C prior to temperature elevation and incubation at 37 or 40˚C (A/B) or preincubated at 37 or 40˚C for 24 h prior to infection and continued incubation at their respective temperatures (C). (A) $TCID_{50}$ titers of supernatants derived from SARS-CoV-2–infected Vero E6 cells incubated at 37˚C over time (h). Means and SD shown. (B/C) $TCID_{50}$ SARS-CoV-2 titers at 24 and 48 h postinfection. Left-hand panel; means and SD. Right-hand panel; black line, median; whisker, 95% confidence interval; all data points shown; *p*-values shown, Mann–Whitney *U* test. (A-C) $N = 3$ independent biological experiments performed in triplicate. Raw values presented in S9 Data. PFU, plaque-forming unit; SARS-CoV-2, Severe Acute Respiratory Syndrome Coronavirus 2.
(TIF)

**S1 Data. RNA-Seq data analysis for Mck-treated respiratory tissues incubated at 37 and 40˚C.** Mck, mock; RNA-Seq, RNA sequencing.
(XLSX)

**S2 Data. Identification of SCV2 SNPs from individual infected respiratory tissues incubated at 37 (37C-1 to 3) or 40˚C (40C-1 to 3).** Sequences were aligned to Wuhan (MN908947.3) as a reference sequence. SCV2, SARS-CoV-2; SNP, single-nucleotide polymorphism.
(XLSX)

**S3 Data. RNA-Seq data analysis for Mck-treated or SCV2-infected respiratory tissues at 37˚C.** Mck, mock; RNA-Seq, RNA sequencing; SCV2, SARS-CoV-2.
(XLSX)

**S4 Data. RNA-Seq data analysis for Mck-treated or SCV2-infected respiratory tissues at 40˚C.** Mck, mock; RNA-Seq, RNA sequencing; SCV2, SARS-CoV-2.
(XLSX)

**S5 Data. RNA-Seq data analysis for Mck-treated or SCV2-infected respiratory tissues at 37˚C and 40˚C, respectively.** Mck, mock; RNA-Seq, RNA sequencing; SCV2, SARS-CoV-2.
(XLSX)

**S6 Data. Comparative DEG analysis of RNA-Seq data derived from Mck-treated or SCV2-infected respiratory tissues at 37 and 40˚C derived from S1, S3, S4, and S5 data analysis.** DEG, differentially expressed gene; Mck, mock; RNA-Seq, RNA sequencing; SCV2, SARS-CoV-2.
(XLSX)

**S7 Data. Comparative lncRNA and miRNA DEG analysis of RNA-Seq data derived from Mck-treated or SCV2-infected respiratory tissues at 37 and 40˚C derived from S1, S3, S4, and S5 data analysis.** DEG, differentially expressed gene; lncRNA, long noncoding RNA; Mck, mock; miRNA, microRNA; RNA-Seq, RNA sequencing; SCV2, SARS-CoV-2.
(XLSX)

**S8 Data. RNA-Seq data analysis for SCV2-infected respiratory tissues at 37 and 40˚C.** RNA-Seq, RNA sequencing; SCV2, SARS-CoV-2.
(XLSX)

**S9 Data. Underlying data used for quantitative analysis in this study.**
(XLSX)

## Acknowledgments

The authors thank Lynn Marion Stevenson, Frazer Bell, and Lynn Oxford (College of Medical, Veterinary and Life Sciences, University of Glasgow) for their exceptional efforts during the UK lockdown. We also thank Matthew Parker (University of Sheffield) for assistance in our periscope analysis.

## Author Contributions

**Conceptualization:** Vanessa Herder, Kieran Dee, Ruth F. Jarrett, Sheila V. Graham, Pablo R. Murcia, Chris Boutell.

**Data curation:** Vanessa Herder, Kieran Dee, Joanna K. Wojtus, Ilaria Epifano, Daniel Gold-farb, Christoforos Rozario, Quan Gu, Ana Da Silva Filipe, Kyriaki Nomikou, Jenna Nichols, Andrew Stevenson, Steven McFarlane, Andreu Masdefiol Garriga, Chris Davis, Jay Allan, Chris Boutell.

**Formal analysis:** Vanessa Herder, Kieran Dee, Joanna K. Wojtus, Ilaria Epifano, Daniel Gold-farb, Christoforos Rozario, Quan Gu, Ana Da Silva Filipe, Kyriaki Nomikou, Jenna Nichols, Ruth F. Jarrett, Andrew Stevenson, Steven McFarlane, Andreu Masdefiol Garriga, Chris Davis, Jay Allan, Chris Boutell.

**Funding acquisition:** Sheila V. Graham, Pablo R. Murcia, Chris Boutell.

**Investigation:** Vanessa Herder, Kieran Dee, Joanna K. Wojtus, Ilaria Epifano, Daniel Gold-farb, Christoforos Rozario, Ana Da Silva Filipe, Kyriaki Nomikou, Jenna Nichols, Andrew Stevenson, Steven McFarlane, Chris Davis, Jay Allan, Chris Boutell.

**Methodology:** Vanessa Herder, Kieran Dee, Joanna K. Wojtus, Ilaria Epifano, Daniel Gold-farb, Christoforos Rozario, Quan Gu, Ana Da Silva Filipe, Kyriaki Nomikou, Jenna Nichols, Ruth F. Jarrett, Andrew Stevenson, Steven McFarlane, Meredith E. Stewart, Agnieszka M. Szemiel, Rute M. Pinto, Chris Davis, Jay Allan, Sheila V. Graham, Pablo R. Murcia, Chris Boutell.

**Project administration:** Vanessa Herder, Kieran Dee, Chris Boutell.

**Resources:** Ruth F. Jarrett, Meredith E. Stewart, Agnieszka M. Szemiel, Rute M. Pinto, Chris Boutell.

**Supervision:** Sheila V. Graham, Pablo R. Murcia, Chris Boutell.

**Validation:** Vanessa Herder, Kieran Dee, Joanna K. Wojtus, Ilaria Epifano, Daniel Goldfarb, Christoforos Rozario, Quan Gu, Ana Da Silva Filipe, Kyriaki Nomikou, Jenna Nichols, Andrew Stevenson, Steven McFarlane, Chris Davis, Jay Allan, Chris Boutell.

**Visualization:** Vanessa Herder, Kieran Dee, Chris Boutell.

**Writing – original draft:** Vanessa Herder, Kieran Dee, Chris Boutell.

**Writing – review & editing:** Vanessa Herder, Kieran Dee, Sheila V. Graham, Pablo R. Murcia, Chris Boutell.

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
