## [Editor Report · Decision Letter 0]

30 Nov 2020

Dear Dr. Boutell, 

Thank you for submitting your manuscript entitled "Elevated temperature inhibits SARS-CoV-2 replication in respiratory epithelium independently of the induction of IFN-mediated innate immune defences" for consideration as a Research Article by PLOS Biology.

Your manuscript has now been evaluated by the PLOS Biology editorial staff as well as by an academic editor with relevant expertise and I am writing to let you know that we would like to send your submission out for external peer review.

Please re-submit your manuscript within two working days, i.e. by Dec 02 2020 11:59PM.

Kind regards,

Paula 

---

Associate Editor

PLOS Biology

---

## [Decision Letter · Decision Letter 1]

6 Jan 2021

Dear Dr. Boutell,

Thank you very much for submitting your manuscript "Elevated temperature inhibits SARS-CoV-2 replication in respiratory epithelium independently of the induction of IFN-mediated innate immune defences" as a Research Article for review by PLOS Biology. As with all papers reviewed by the journal, yours was assessed and discussed by the PLOS Biology editors, by an academic editor with relevant expertise, and by several independent reviewers. Based on the reviews, I regret that we will not be pursuing this manuscript for publication in the journal.

Both referees agree have concerns with the results and conclusions given that the cells are from only one donor. Reviewer #1 is not convinced that the restriction is independent of IFN and asks you to show that the virus productively infects the cells in the absence of the immune response. Reviewer #2 also asks you to show at what stage of the viral life cycle the restriction takes place, add controls of cell viability, and sequencing outcoming virus. Both reviewers find the coronavirus restriction at higher temperatures interesting but the high amount of work needed in the revision in order to address the reviewers' concerns precludes publication in PLOS Biology at this stage. 

The reviews are attached, and we hope they may help you should you decide to revise the manuscript for submission elsewhere. I am sorry that we cannot be more positive on this occasion. 

I hope you appreciate the reasons for this decision and will consider PLOS Biology for other submissions in the future. Thank you for your support of PLOS and of Open Access publishing.

Sincerely,

Paula Jauregui

---

Associate Editor,

pjaureguionieva@plos.org,

PLOS Biology

Reviewer expertise:

Reviewer #1: Host - pathogen interactions in respiratory epithelium.

Reviewer #2: Coronavirus

Reviewer remarks:

Reviewer #1: The manuscript by Vanessa Herder, Kieran Dee, and colleagues demonstrates that SARS-CoV-2 replication is constrained at ≥39°C using different cell culture models. The results complement the increasing body of knowledge on SARS-CoV-2 replication in the human respiratory epithelium that likely can be extended to other pathogens and is therefore of potential interest to a broad audience. However, certain claims made by the authors are not completely supported by the data at hand and need to be further substantiated by additional experimental data, which I would like to clarify below.

Major:

One of the most outstanding issues with the manuscript is that throughout the manuscript a large body of the data is based on invalid experimental design. For instance, the differential gene expression analysis is questionable as the RNA-seq analysis has unfortunately been performed on three technical replicates per condition (Lines 767 - 768, 783 - 784, 817 - 818, 836 - 837, 878 - 879, 891 - 892, 908 - 909, and 927 - 928), instead of three or more independent biological replicates per condition. Thus, the biological number of independent experiments equals one, and therefore, the data does not represent the biological variance, but rather the variance within a single sample. This should be resolved by complementing the existing data with additional biological replicates. In addition to the RNA-seq data, the statistical analysis on the viral replication kinetics has been done using the technical replicates (e.g N=300, N=5, N=11 or N=9) rather than biological replicates (N=3) (line 743, 792 - 793, 797, 800 - 801, 847 - 848). Please clarify this and revise the statistical analysis throughout the manuscript accordingly. 

Despite the aforementioned discrepancies, this reviewer acknowledges that the authors elegantly demonstrate on different levels that an elevated temperature (39°C/40°C) alters the in vitro respiratory epithelium and constrains SARS-CoV-2 replication in different cell culture models. However, the reviewer is not convinced by the claim that restriction of SARS-CoV-2 is independent of the induction of type-I and type-III IFN-mediated antiviral immune defenses. This as the authors do not directly and conclusively demonstrate that SARS-CoV-2 (i) enters the respiratory epithelium and (ii) that there is robust productive viral replication in (iii) the absence of an innate immune response. This is for instance exemplified by the results that there are no antigen-positive cells for SARS-CoV-2 detected at 40°C (Figure 8E). Moreover, it cannot be excluded that the low concentration of viral RNA detected at 40°C reflects residual virus from the inoculum (which is missing from the figures), as the apical surface is only washed once after a 2 hour inoculation (Line 375 - 378). In line with this, the viral replication kinetics (TCID50) at 72 hours are all likely in proximity to or below the limit of detection of the assay and might reflect residual virus from the inoculum.

Therefore, in order to support the major claim that "the restriction of SARS-CoV-2 is independent of the induction of type-I and type-III IFN-mediated antiviral immune defenses", the reviewer would like to ask the authors to provide the following experimental data: 

1. Validation that SARS-CoV-2 enters and productively infects the cultures, by using a PCR specifically targeting the leader-body junction of a sub-genomic mRNA of SARS-CoV-2. 

2. Monitoring of the viral kinetics and host response in an experiment whereby during the course of the experiment the temperature is being elevated from the core body temperature (37°C) to 40°C in the absence or presence of the Janus-associated kinase (JAK) inhibitor Ruxolitinib.

Minor:

Please include the viral inoculum in all of the replication kinetic figures throughout the manuscript.

Figure1B and E, why is there a drop in viral titer at 120 hpi?

Line 353 "VeroE6" should be written as "Vero E6"

Figure 5B please include a color legend for the Chord diagram, in either the figure or figure legend.

Reviewer #2: The manuscript by Herder et al. studies the effect of elevated temperature on SARS-CoV-2 replication. Elevated temperature mimics the fever, which is one of the most frequent symptoms of SARS-CoV-2 induced disease (COVID-19). 

Using an experimental model that consists in differentiated primary human bronchial 3D epithelial (HBE) cells grown under air-liquid interface (ALI), the replication of SARS-CoV-2 and the cellular gene expression profile were analyzed at the physiological temperature of 37ºC and at the hyperthermic temperatures of 39ºC and 40ºC. RNAseq results showed that temperature elevation induced a heat stress response in the respiratory tissue model. It also restricted the replication of SARS-CoV-2, as confirmed by RNAseq and titration of infectious virus. The significant reduction of virus replication at 40ºC failed to induce the production of interferon. 

The experimental model and the design of the experiments are appropriate. However, the novelty of the results is limited. As mentioned in the manuscript, the analysis of the cellular pathways differentially expressed in response to SARS-CoV-2 infection (Fig. 2) has been previously reported (references 16, 18-21, 30-32). It has also been extensively described that the temperature elevation induces in tissues a heat stress response (Fig. 3). 

The most novel aspect of the manuscript is the description of reduced viral RNA levels at elevated temperatures, which leads to a reduction in viral titers (around 2-log units lower at 40ºC than at 37ºC). This effect has been previously described in influenza virus as a result of decreased thermal stability of the RNA-dependent RNA polymerase. However, in coronaviruses, the impact of temperature elevation has not been analyzed in depth. The mouse hepatitis coronavirus replicates to high titers both at 37º and 40°C, whereas a temperature-sensitive mutant with mutations in the nsp3 showed replication defects at 40ºC. The relevant observation that the replication of a clinical isolate of SARS-CoV-2 is sensitive to temperature would require further research to characterize the mechanisms responsible for this sensitivity. 

Specific points

1. Primary human bronchial epithelial cells from one single human donor were used for the experiments. Because of the high inter-donor variability, samples from several donors should have been used in order to achieve more representative and conclusive results.

2. The impact of the higher temperatures in the viability of the cells should be analyzed to differentiate whether the restriction of viral replication is a direct effect on the activity of viral proteins or an indirect effect mediated by a reduced cell viability. 

3. Are there any mutations in consensus sequence of the virus after the high-temperature incubation period?

4. The mechanisms involved in the reduced viral growth should be studied by analyzing whether they are acting at the level of entry, viral RNA synthesis, protein synthesis, morphogenesis…

Minor points

5. Fig. 1B and 1C graphs showing all data points or means and SD, respectively, are redundant and should be unified. The same is true for Fig. 4A-B and Fig. for 4C-D.

6. Fig. 1. The amount of virus inoculum in infections should be indicated as a multiplicity of infection (MOI=PFU per cell) instead of PFU per transwell.

---

## [Editor Report · Decision Letter 2]

21 Jul 2021

Dear Dr. Boutell,

Thank you very much for your appeal related to the manuscript "Elevated temperature inhibits SARS-CoV-2 replication in respiratory epithelium independently of the induction of IFN-mediated innate immune defences" for consideration at PLOS Biology. Your appeal has been evaluated by the PLOS Biology editors, and an Academic Editor with relevant expertise. 

After discussing about it, we would welcome re-submission of a much-revised version that takes into account the previous reviewers' comments. Your revised manuscript will be sent for further evaluation by the reviewers.

We expect to receive your revised manuscript within 3 months. 

**IMPORTANT - SUBMITTING YOUR REVISION**

*Re-submission Checklist*

*Published Peer Review*

*PLOS Data Policy*

*Blot and Gel Data Policy*

Sincerely,

Paula 

---

Paula Jauregui, PhD

Associate Editor

PLOS Biology

---

## [Decision Letter · Decision Letter 3]

23 Sep 2021

Dear Dr Boutell,

Thank you for submitting a revised version of your manuscript "Elevated temperature inhibits SARS-CoV-2 replication in respiratory epithelium independently of IFN-mediated immune defences" for consideration as a Research Article at PLOS Biology. I have taken over its handling during the absence of my colleague Paula Jauregui from the office, in order to prevent unnecessary loss of time. This revised version of your manuscript has been evaluated by the PLOS Biology editors, the Academic Editor and the original reviewers.

As you will see below, although both reviewers acknowledge the extensive revisions made, they have some outstanding concerns that we consider would need to be satisfactorily addressed for us to publish the work. Reviewer 1 raises a significant concern with the statistics and some data presentation, and urges you to use additional donors for the RNA-seq analysis - a point the Academic Editor thought particularly important to address. Reviewer 2 points out an error in the normalization of sgmRNA levels and requests the data in fig 7E be reanalysed with this in mind.

Ideally, if we were to publish the work, we would do so in this calendar year, as we published a study with similar conclusions earlier in the year and it would be in your best interest to have the same publication year. Of course, the data would need to be solid for us to move to acceptance. 

We have thus set your revision time to 6 weeks, hoping these issues can be addressed in a reasonable time frame, as most underlying data is available. We will then assess your revised manuscript and your response to the reviewers' comments and we may consult the reviewers again.

Please email us (plosbiology@plos.org) if you have any questions or concerns. At this stage, your manuscript remains formally under active consideration at our journal; please notify us by email if you do not intend to submit a revision so that we may end consideration of the manuscript at PLOS Biology.

**IMPORTANT - SUBMITTING YOUR REVISION**

*Resubmission Checklist*

*Published Peer Review*

*PLOS Data Policy*

*Blot and Gel Data Policy*

Sincerely,

Nonia

Nonia Pariente, PhD

Editor in Chief

PLOS Biology

on behalf of 

Paula Jauregui, PhD

Editor

PLOS Biology

REVIEWS:

Reviewer #1: The reviewer acknowledges and values the extensive revision that is presented by Vanessa Herder, Kieran Dee, and colleagues. This has increased the overall study, and thereby reviewer is still convinced that the concept is of interest to a broad scientific audience. However, the reviewer regrets that the current way of data presentation and statistical testing are inappropriate to validate whether the additional experimental data confirms the earlier observations made with the RNA-seq analysis, which is regrettably still a single independent biological donor. This crucial point requires specific attention from the authors and is described in more detail below. 

Major:

The reviewer would like to thank the authors for the detailed explanation on how the statistical analysis was conducted and hopes that the commercial multipotent basal airway cells to establish differentiated airway epithelial cell cultures are not contaminated with fibroblasts. However, as indicated above, the reviewer unfortunately strongly disagrees on the appropriateness of the statistical testing and underlying scientific reasoning of the authors. This because measurement of individual wells from the same biological donor, differentiated in parallel from the same batch of cells and used at the same instance, need to be considered as repeated measurements of the noise (e.g., cellular heterogeneity) associated with the plastics, culture reagents and handling procedures. Irrespective whether this is conventional or more advanced primary cell culture. In a more simplified analogy, one cannot consider a single 96-well cluster plate with Vero E6 cells obtained from a single culture flask, that are all infected with SARS-CoV-2 at the same time as 96 individual biological replicates. Instead, this is a single independent biological replicate (N=1) with 96 technical replicates, which all have an inherent cellular heterogeneity (e.g., distinct receptor expression profiles, cell cycle, etc.). Hence, if primary airway cells of a single biological donor originating from a culture flask are divided over different transwell inserts, these individual inserts from the same batch and donor need to be considered as technical replicates, as they do not represent biological variation.

Therefore, the reviewer urges the authors to include more donors for the RNA-seq analysis (such as from the samples included in the revision), and stresses that the authors update their data and statistical analysis and presentation throughout the manuscript. This to reflect the actual biological variation from the 3 different donors used in the manuscript, rather than stratifying the data by donor and performing statistical analysis on technical replicates for each individual donor (e.g., N=9 in Fig3, 4A-C, and Fig 6E-F). This, as the latter needs to be considered as "pseudo-replication", the inflation of the actual number of samples, which does not reflect the actual biological variation the authors would like to interpret their data from, nor want to convey to the scientific audience.

The graphs in Fig 6E-D, are not intuitive to understand and need to be reanalysed using the more appropriate delta-delta Ct approach (https://doi.org/10.1006/meth.2001.1262), and should reflect the actual biological variation. 

The statement in line 147 - 148 is incorrect, as it has previously been shown that temperature restricts SARS-CoV-2 replication, as the authors themselves also highlight in the introduction Line 68 - 71. This should be correct to for instance: "To our knowledge, these data for the first time demonstrate that febrile temperatures restrict SARS-CoV-2 replication…."

Minor:

- It does not make sense to abbreviate the well-established abbreviation SARS-CoV-2. 

- Line 397; 1.1x10E5 cells/well should this not be ten-fold lower?

Reviewer #2: Most of the points raised by this reviewer have been satisfactorily addressed in the revised version of the manuscript by Herder et al.

However, there is one new point that should be clarified. It is claimed in the Abstract that temperatures >39ºC restrict viral growth by limiting viral transcription. This statement is supported by the RNA-seq results shown in Fig. 7D and 7E. 

RNA synthesis in coronaviruses include two differentiated processes, replication, which copies the whole genomic RNA (gRNA), and transcription, which synthesizes subgenomic mRNAs (sgmRNAs) of 3' end genes by a discontinuous mechanism. Regulation of sgmRNA levels does primarily depends on the levels of genomic RNA acting as templates for transcription, in addition to specific transcription-regulating factors.

Then, for sgmRNA quantification, the read counts of sgmRNAs should be normalized to the levels of gRNA in order to analyze whether or not lower levels of sgmRNAs are just a direct consequence of reduced gRNA levels. That being the case, a specific regulatory effect of temperature on transcription could not be confirmed.

Therefore, sgmRNAs (Fig. 7E) should be normalized to gRNA levels, which are determined from the read counts of Orf1ab, only produced by replication and not by transcription. In fact, including Orf1a in Fig. 7E as a sgmRNA is not correct, since Orf1a is not synthesized by the transcription mechanism. 

In summary, to conclude an effect of tissue temperature specifically on viral transcription, a reduction on sgmRNA levels, further than that derived from the reduction in template gRNA levels should be confirmed.

---

## [Editor Report · Decision Letter 4]

25 Nov 2021

Dear Dr. Boutell,

Thank you for submitting your revised Research Article entitled "Elevated temperature inhibits SARS-CoV-2 replication in respiratory epithelium independently of IFN-mediated innate immune defences" for publication in PLOS Biology. I have now discussed your revision with the Academic Editor. 

We will probably accept this manuscript for publication, provided you satisfactorily address the following data and other policy-related requests.

DATA POLICY:

2) Deposition in a publicly available repository. **Please also provide the accession code or a reviewer link so that we may view your data before publication. **

Regardless of the method selected, please ensure that you provide the individual numerical values that underlie the summary data displayed in the following figure panels as they are essential for readers to assess your analysis and to reproduce it: Figures 1CD, 2BCDEFG, 3ABC, 4ABC, 5ABC, 6ABCDEFG, 7ABCDEG, S1ABC, S2ABCD, S3ABC, S4ABC, S5ABCD, S6AB, S7ABC

**Please also ensure that figure legends in your manuscript include information on where the underlying data can be found, and ensure your supplemental data file/s has a legend.**

**Please ensure that your Data Statement in the submission system accurately describes where your data can be found.**

Please provide a blurb which (if accepted) will be included in our weekly and monthly Electronic Table of Contents, sent out to readers of PLOS Biology, and may be used to promote your article in social media. The blurb should be about 30-40 words long and is subject to editorial changes. It should, without exaggeration, entice people to read your manuscript. It should not be redundant with the title and should not contain acronyms or abbreviations. Please add it in the requested place in our system.

We expect to receive your revised manuscript within 1 week.

*Published Peer Review History*

*Early Version*

Sincerely,

Paula

---

Associate Editor,

pjaureguionieva@plos.org,

PLOS Biology

---

## [Editor Report · Decision Letter 5]

3 Dec 2021

Dear Dr. Boutell,

On behalf of my colleagues and the Academic Editor, Andrea Cimarelli, I am pleased to say that we can in principle accept your Research Article "Elevated temperature inhibits SARS-CoV-2 replication in respiratory epithelium independently of IFN-mediated innate immune defences" for publication in PLOS Biology, provided you address any remaining formatting and reporting issues. These will be detailed in an email that will follow this letter and that you will usually receive within 2-3 business days, during which time no action is required from you. Please note that we will not be able to formally accept your manuscript and schedule it for publication until you have any requested changes.

PRESS

Sincerely, 

Paula 

---

Paula Jauregui, PhD 

Associate Editor 

PLOS Biology
